# RLRF: Competitive Search Agent Design via Reinforcement Learning from Ranker Feedback

## Abstract

Competitive search is a setting where document publishers modify them to improve their ranking in response to a query. Recently, publishers have increasingly leveraged LLMs to generate and modify competitive content. We introduce Reinforcement Learning from Ranker Feedback (RLRF), a framework that trains LLMs using preference datasets derived from ranking competitions. The goal of a publisher (LLM-based) agent is to optimize content for improved ranking while accounting for the strategies of competing agents. We generate the datasets using approaches that do not rely on human-authored data. We show that our proposed agents consistently and substantially outperform previously suggested approaches for LLM-based competitive document modification. We further show that our agents are effective with ranking functions they were not trained for (i.e., out of distribution) and they adapt to strategic opponents. These findings provide support to the significant potential of using reinforcement learning in competitive search.

## 1 Introduction

*Competitive Search* refers to a search setting where strategic document authors actively optimize their documents' content to improve ranking in response to a query induced by a search engine (Kurland & Tennenholtz, 2022). Ranking competitions are particularly intense in commercial domains, where a higher search rank directly translates into increased traffic, influence, and revenue (Joachims et al., 2017). As search algorithms evolve, so do the modifications applied by publishers, making competitive search a dynamic interplay between the search algorithms and strategic content creation.

While traditional publishers' strategies often relied on surface-level techniques such as keyword stuffing (designed to exploit the bag-of-words nature of early search algorithms; Zuze & Weideman, 2013; Drivas et al., 2017) or non-content-based approaches (aimed at manipulating PageRank-based systems; Alice, 2006; Bar-Ilan, 2007), the rise of large language models (LLMs) has fundamentally reshaped the competitive search landscape. Modern search engines increasingly rely on advanced neural ranking methods such as dense retrieval[1], which prioritize semantic understanding over exact keyword matches (Zhao et al., 2024b). As a result, publishers now focus on crafting content that aligns with the deeper meaning and intent behind user queries.

At the same time, the rise of LLMs has made it easier for publishers to engage in this new form of semantically driven optimization. LLMs not only excel in core natural language processing tasks such as sentiment analysis and text generation (Brown et al., 2020; Peng et al., 2023; Zhang et al., 2023; Susnjak, 2024; Wang et al., 2024d), but also in competitive tasks that require strategic reasoning (Shapira et al., 2024b; Raman et al., 2024; Akata et al., 2025), positioning them as powerful tools for navigating the increasingly complex and competitive search ecosystem. This dual role of LLMs, as both a force in ranking algorithms and a content creation tool for publishers, has created a new era of competitive search centered on strategic content design (Nachimovsky et al., 2025).

The role of LLM-based agents as strategic publishers in competitive search environments has not yet been systematically studied. Mordo et al. (2025a) introduced a simulation framework that models ranking competitions involving both human and LLM-based participants. Bardas et al.

---

[1]Dense retrieval refers to a retrieval paradigm in which both queries and documents are encoded into dense vector representations (typically using neural networks), and relevance is estimated via vector similarity (e.g., dot product), rather than sparse term overlap as in traditional methods (Zhao et al., 2024b).

(2025) employed a framework to evaluate the effectiveness of LLM agents in one-shot competitive search settings under different prompting and feedback strategies. This raises a natural question: **Can LLM-based strategic agents be improved beyond prompting by training them — using reinforcement learning — to optimize for ranking competition objectives, i.e., to be ranked as highly as possible during the competition?**

In this work, we introduce a novel paradigm for training LLM-based agents in competitive search environment that leverages reinforcement learning (RL) alignment techniques to improve the content produced by agents in terms of rankings. The key idea is to align the LLMs using feedback induced from the ranker's output (i.e., the ranking), where this feedback is reformulated as prompts for the LLM-based agents. By incorporating this feedback, the agent learns to produce content that is more likely to be ranked higher across a variety of queries and competitive contexts. Importantly, the RL-based alignment occurs only at training time; at test time, the agents operate solely through prompting, without additional optimization. We refer to this approach as *Reinforcement Learning from Ranker Feedback (RLRF)*. Agents trained using this paradigm are henceforth referred to as RL-aligned agents or **RA agents** in short. Our contributions are as follows:

- We formalize the setting of *competitive search* as a learning problem in which LLM-based agents generate content to maximize their rank in a dynamic ranking environment.
- We introduce the novel RLRF methodology, which aligns the LLM with the competitive ranking objective. We characterize two key aspects of the learning process of RA agents: (i) aligning with the search engine's ranking function, and (ii) adapting to strategic opponents in a ranking competition.
- We train our agent on synthetic datasets generated using two approaches: Static Generation (SG), which produces documents' modifications independent of other agents, and Dynamic Generation (DG), which simulates multi-agent competition.
- We demonstrate the effectiveness of RLRF through extensive experiments in a controlled competitive search framework, showing that agents aligned with RLRF consistently outperform baseline prompting-based approaches across a range of queries and competitive settings.
- We show that RA agents trained with one ranker can transfer effectively to different ranking functions.

## 2 RELATED WORK

**Game-theoretic Foundations of Competitive Search** There is a growing body of work on competitive search settings where document authors modify their documents so as to improve their future ranking in response to queries (Kurland & Tennenholtz, 2022). Specifically, game theoretic approaches were used, alongside empirical studies, to analyze ranking paradigms (Kurland & Tennenholtz, 2022; Ben Basat et al., 2015; 2017; Ben-Porat et al., 2019; Nachimovsky et al., 2024; Mordo et al., 2025b) (e.g., whether they lead to equilibrium), to study authors' document modification strategies (Raifer et al., 2017; Ben-Porat et al., 2019; Madmon et al., 2025a;b), and to explore potential corpus-based enrichment approaches to ensure equilibrium (Nachimovsky & Tennenholtz, 2025). In contrast, we focus on RL-based training of LLM agents that act as document authors.

**LLMs in Competitive Environments** LLMs have recently shown strong potential as rational agents in strategic interactions (Xi et al., 2023; Fu et al., 2023; Wang et al., 2024a; Guo et al., 2024a;b; Akata et al., 2025; Xie et al., 2025). Recent benchmarks were used to evaluate LLM performance in complex multi-agent decision-making tasks, assessing both individual rationality (Raman et al., 2024; 2025) and collective economic measures such as efficiency and fairness (Shapira et al., 2024b). One of the promising directions is simulating competitive tasks using LLMs (Zhao et al., 2024a); the theoretical aspects are sometimes analyzed using game theoretic models (Mao et al., 2024). As highlighted by Nachimovsky et al. (2025), LLMs can play different roles in the competitive search ecosystem. While most of the previous work focused on the ranker's perspective (Gao et al., 2024c; Wang et al., 2024c; Rathee et al., 2025; Guo et al., 2025b), we focus on utilizing LLMs to generate documents from the perspective of the (strategic) publisher.

Bardas et al. (2025) initiated the study of LLM-based agents, showing that few-shot LLMs can perform on par with human publishers in a single-round ranking promotion setting. In contrast, our work addresses a more complex and practical framework where agents modify their content across long-term interactions with other agents. Building on the competitive search simulation framework of

Mordo et al. (2025a), we show that RLRF techniques can enhance LLM-based agents to outperform the few-shot agents of Bardas et al. (2025).

**RL in Competitive Settings** RL has long been used to train agents in competitive and multi-agent environments, achieving remarkable success in board and video games (Vinyals et al., 2017; Xenou et al., 2018; Vinyals et al., 2019; Li et al., 2024). More recently, RL from human feedback (RLHF) has emerged as a key technique for aligning large language models (LLMs) with human preferences in non-strategic tasks such as summarization and dialogue generation (Christiano et al., 2017; Ouyang et al., 2022; Shen et al., 2023; Gao et al., 2024b; Tennenholtz et al., 2024). To scale this approach, RL from AI feedback (RLAIF) has been proposed, replacing human evaluators with LLM-based feedback to improve scalability (Bai et al., 2022b; Lee et al., 2024). Subsequent work applied RL-based techniques to enhance the decision-making abilities of LLMs (Schmied et al., 2025) and to optimize content generation in competitive landscapes (Sharma et al., 2022; Coppolillo et al., 2024). RL has also been applied to recommendation systems to improve recommendation performance by optimizing long-term user engagement (Sun et al., 2024). More recently, Ye et al. (2025) introduced an RL-based generator agent that strategically uploads items into recommender environments. While both works use LLM-based agents to generate content, their focus is on simulating generators to evaluate recommender systems and on aligning synthetic data with real-world distributions (e.g., YouTube). In contrast, our goal is to design long-term strategies for agents in multi-agent settings rather than to evaluate recommenders.

**RL in Information Retrieval** An RL-based relevance feedback approach improved retrieval effectiveness by iteratively adapting to user interactions (Montazeralghaem et al., 2020) (a.k.a., dynamic retrieval (Yang et al., 2016)). RL was also used with LLMs to guide interaction with search engines (Jin et al., 2025) and to enhance query generation and expansion (Jiang et al., 2025; Yang et al., 2025). In contrast to this line of work which focuses on the ranker, our focus is on content creation by publishers aiming to improve the ranking of their documents.

## 3 TASK DEFINITION AND APPROACH

We address the task of designing a document authoring agent which competes in a repeated ranking game (Kurland & Tennenholtz, 2022). In each game, a fixed set of agents repeatedly compete for the highest ranking induced by an undisclosed ranking function for a given query. A competition consists of multiple games, where each game is associated with a distinct query. Each game lasts for several rounds. At the beginning of a game, each agent is assigned with an identical initial document. From the second round onward, all agents simultaneously modify their documents based on the ranking in the previous rounds. After all agents submit the modified versions of their documents, the system applies a non-disclosed ranking function; specifically, only the ordering of documents is provided every round. The goal of each agent is to strategically adapt its document over the course of a game in order to consistently achieve high ranks. A schematic illustration of a single game is shown in Figure 3 in Appendix A.

**Learning Approach** We employ **Reinforcement Learning from Ranking Feedback (RLRF)** to train our agent, henceforth referred to as **RL-aligned agent (RA agent)**. Specifically, the LLM is trained with signals derived from rankings, enabling it to perform more effectively in ranking competitions at test time. To this end, we generate synthetic data to construct a preference dataset[2] and train the agent to increase the likelihood of content modifications that lead to higher ranks while decreasing the likelihood of those that result in lower ranks. The algorithms implementing RLRF using DPO[3] (Rafailov et al., 2024) are presented in Figure 1. The key difference between the two algorithms lies in how the documents are generated. In the **Static Generation (SG; Algorithm 1)** setting, for each query, an LLM first generates a pseudo-relevant document to the query, independent of any competitive context. Based on this document, multiple modified variants are then generated using

---

[2]A preference dataset consists of triplets: (i) a prompt or feedback context, (ii) a positive example (a document modification that is ranked above another candidate), and (iii) a negative example (the lower-ranked candidate); positive/negative labels are derived from the ranker's ordering.

[3]The choice of the DPO algorithm over alternative methods is discussed in Section 4.3.

---

**Algorithm 1** RLRF Agent: Static Generation

**Require:** LLM $M$, queries $Q_{train}$, ranker $R$
**Ensure:** Fine-tuned agent $M^*$
**Ensure:** Initialize the preference dataset
1: **for** each $q \in Q_{train}$ **do**
2:     Generate a pseudo-relevant document for $q$ with a prompt[a]
3:     The agent modifies $N$ times its document with a prompt[b]
4:     Ranker $R$ ranks the $N$ modified documents
5:     Add to preference dataset: $(prompt, d_{top}, d_{bottom})$ where $d_{top}$ and $d_{down}$ are the highest and lowest ranked documents, respectively.
6: **end for**
7: Update $M$ using the preference dataset with the **DPO** algorithm
8: **return** $M^*$

---

[a]See Appendix B.1 Figure 4.
[b]See Appendix B.1 Figure 5.

---

**Algorithm 2** RLRF Agent: Dynamic Generation

**Require:** LLM $M$, queries $Q_{train}$, ranker $R$, set of rounds $T$
**Ensure:** Fine-tuned agent $M^*$
**Ensure:** Initialize the preference dataset
1: **for** each $q \in Q_{train}$ **do**
2:     Initialize a ranking competition with an initial document
3:     **for** each round $t \in T$ **do**
4:         Every agent modifies its document with a prompt[a] (Bardas et al., 2025)
5:         Ranker $R$ ranks all documents
6:         Add to preference dataset: $(prompt, d_{top}, d_{bottom})$ where $d_{top}$ and $d_{down}$ are the highest and lowest ranked documents (from the ranker's output) respectively.
7:     **end for**
8: **end for**
9: Update $M$ using the preference dataset with **DPO** algorithm
10: **return** $M^*$

---

[a]See Section 4.1.

Figure 1: RLRF Agent Designs: Static (left) vs. Dynamic (right).

prompts that instruct the LLM to revise the document in different ways[4]. A ranking is induced over the resulting pool of documents, and the preference dataset is extracted from the highest- and lowest-ranked variants. This approach enables learning how document modifications influence rankings. In contrast, in the **Dynamic Generation (DG; Algorithm 2)** setting, there is a repeated ranking game, where multiple instances of the same LLM iteratively modify their documents in response to rankings. In this setup, the data generation procedure produces a preference dataset that reflects the evolving competitive dynamics across rounds. Consequently, during training, the algorithm aligns the agent not only with the ranker's preferences but also with the document-modification strategies that emerge over time in the competition. This alignment enables the agent to adapt its document-modification strategy across rounds and achieve improved performance, as we show in Section 5. Importantly, our core novelty lies in training an agent for an unknown ranker using only implicit signals induced from rankings, while simultaneously accounting for the strategic behavior of other agents in the competition. Additional algorithmic details are provided in Appendix C. Preliminaries on RL and DPO are provided in Appendix A.

# 4 EXPERIMENTAL SETTING

In this section, we detail the framework employed for training and evaluating the RA agent. The agent is trained on the synthetic preference datasets derived from a simulated ranking competition between large language models (LLMs). To train our agent, we adopt Direct Preference Optimization (DPO; Rafailov et al., 2024), utilizing a set of prompts introduced in prior work (Bardas et al., 2025). The performance of the resulting agent is evaluated using the LEMSS simulated environment for LLM-based ranking competitions (Mordo et al., 2025a).

## 4.1 COMPONENTS

**LLMs and Prompts**   We used lightweight instruct-tuned language models (< 10B parameters) as our agents: Llama3.1 (Dubey et al., 2024), Mistral (Jiang et al., 2023), Gemma2 (Gemma Team et al.), and Qwen2.5[5] (Qwen et al., 2024). The choice of LLMs was motivated by two reasons. First, using lightweight models allows us to conduct large-scale training and evaluation under reasonable

---

[4]The prompts are presented in Appendix B.1
[5]Models sourced from the Hugging Face repository: meta-llama/Meta-Llama-3.1-8B-Instruct, mistralai/Mistral-8B-Instruct-2410, google/gemma-2-9b, and Qwen/Qwen2.5-7B-Instruct.

computational constraints (Belcak et al., 2025). Second, this setup aligns, and therefore allows comparison, with prior work on competitive search (Mordo et al., 2025a), where models with up to 10 billion parameters were used to ensure reproducibility and accessibility (Belcak et al., 2025). The prompts used in our experiments are from Bardas et al. (2025); LLM-based agents guided by these prompts consistently outperformed student participants in single-round document modification. Specifically, we employ (i) the Pairwise Prompt agent (**PAW**) and (ii) the Listwise Prompt agent (**LSW**) (Bardas et al., 2025). The PAW prompt consists of the last three rounds of a pair of documents and their ranks with respect to the query. The LSW prompt consists of the last two rounds of the entire ranked list with respect to the query. We denote these prompt-based agents as **non-aligned agents (NA agents)**, since they were not trained prior to the ranking competition but rather calibrated only through hyper-parameter tuning and prompt engineering.

**Ranking Functions** We employed three dense retrieval ranking functions and one sparse retrieval method. The dense rankers, following prior work on ranking competitions (Mordo et al., 2025a), are: E5 in both its unsupervised and supervised variants (Wang et al., 2024b), and Contriever[6] (Izacard et al., 2022). The sparse ranker is Okapi BM25 (Robertson et al., 1993). For the dense retrieval models, ranking scores for document–query pairs were computed using cosine similarity between their respective embedding vectors. For the BM25 ranking function, we extracted inverse document frequency (IDF) features from a 59,000-document subset of the English Wikipedia, based on a 2020 dump. The text was normalized using Krovetz stemming, following the pre-processing protocol described in Frej et al. (2020a;b).

**Queries and Initial Documents** Each game is assigned with a query for which the agents compete. The game begins with the same initial document that each agent is required to modify in an effort to improve its ranking for the given query. We selected 500 queries from the Passage Ranking task of the TREC 2022 test collection, which is based on the MS MARCO dataset (Payal Bajaj et al., 2016; Craswell et al., 2025); the queries were divided randomly to 90% for the training dataset and 10% for the test dataset. For each query, we also selected an initial document from the MS MARCO Passage collection that had been manually judged as highly relevant to that query[7]. The documents are therefore short as in prior studies of competitive search (Raifer et al., 2017).

## 4.2 DATA GENERATION

Recent work demonstrated remarkable success in improving the performance of AI models using synthetic data in strategic decision-making (Shapira et al., 2024a; 2025) and gaming scenarios (Silver et al., 2017; 2018). Inspired by this line of research, we constructed synthetic datasets to train and optimize LLM-based agents in our competitive search setting. In alignment with real-world scenarios, where Web publishers typically do not have knowledge of the internals of ranking algorithms, we assume that agents are exposed only to the ranked list of documents. The use of generative AI to construct preference datasets tailored to task-specific fine-tuning of language models has been studied in prior work (Bai et al., 2022a; Lee et al., 2024; Gao et al., 2024a). Inspired by this line of research, we generate training data by sampling outputs from a ranker using two methods: Static Generation (SG) and Dynamic Generation (DG) as discussed in Section 3. More technical details are provided in Appendices B and C.1.

## 4.3 AGENT TRAINING

We train the RA agents using the data generation methods introduced in Section 4.2. In line with prior work on competitive search, we instruct the agents to generate short documents of approximately 150 words (Bardas et al., 2025; Mordo et al., 2025a). In contrast to RLHF (Christiano et al., 2017), which aligns model outputs with human preferences, our objective is to align agent behavior (namely, document modification strategies) with the preferences of a ranker. Importantly, the agent is only exposed to rankings for a limited set of queries, without access to scores or model internals. Rather than relying on less stable optimization methods such as Proximal Policy Optimization (PPO; Schulman et al., 2017), which typically require training an explicit reward model and collecting a

---

[6]The dense models were obtained from the Hugging Face repository: intfloat/e5-large-unsupervised, intfloat/e5-large-supervised, and facebook/contriever.

[7]Three out of three annotators judged the document as relevant to the query.

large dataset to approximate the behavior of a ranker, we adopt Direct Preference Optimization[8] (DPO; Rafailov et al., 2024). DPO offers a more stable and sample-efficient alternative, as it directly optimizes model parameters using pairwise preference data (Wu et al., 2023; Rafailov et al., 2024). As a result, DPO is easier to tune, and has lower optimization complexity — an important consideration in our multi-agent simulation setup. We also note that our goal in this work is to provide a proof of concept demonstrating that RL-style preference optimization can improve content generation in ranking competitions, rather than to identify the globally optimal RL algorithm. Given our limited computational resources, we prioritized a stable and lightweight method, and we acknowledge that in some settings carefully tuned PPO can outperform DPO.

## 4.4 EVALUATION

Our setting models repeated interactions where agents iteratively modify their documents over multiple rounds in response to ranking and the strategic behavior of other agents. We present two evaluation settings: **Homogeneous (denoted Ho) and Heterogeneous (denoted He)**. In the Ho setting the RA agent competes against duplications of NA agents (non-aligned agents) with the same language models as the RA agent. In the He setting the RA agent competes against NA agents with different language models. Recall that the feedback to all the agents is provided by using the LSW or the PAW prompts (Bardas et al., 2025). For each setting, we compare the **win-rate**[9] of the RA agent against the *best* performing NA agent for that specific setting[10]. We evaluate an agent performance in the ranking competition simulated using LEMSS (Mordo et al., 2025a) measuring the win-rate averaged across games in the competition. We also define a *random baseline* whose performance is the expected win-rate if all agents have an equal probability of winning each round (i.e., $1/k$ for $k$ competing agents). See Appendix D for detailed description of the measures. Statistical significance is measured using a two-tailed paired permutation test with $p = 0.05$ and $10,000$ permutations.

In addition to win-rate, we evaluate the **faithfulness** of the modified documents to their original counterparts in order to capture cases of substantial modifications made in pursuit of ranking promotion. Following Bardas et al. (2025), we employ an NLI model developed by Gekhman et al. (2023) to compute whether a modified document is entailed by the initial document. A formal definition of this measure is provided in Appendix D.

## 5 ANALYSIS AND RESULTS

We begin by presenting the research questions (RQs) that guide the evaluation of the RA agents. For each RQ, we define one or more experimental settings that enable a comprehensive analysis of the agent's behavior and performance:

- **RQ1:** To what extent does the RA agent outperform NA agents in repeated ranking competitions between LLMs?
- **RQ2:** How well does the RA agent generalize to unseen ranking functions, and how robust is it to potential misalignment between training and test-time ranking functions?

We evaluate the RA agent (compared to NA agent) in simulated ranking competitions. For RQ1 we use the two configurations Ho and He. For RQ2, we used the He setup, as it is considered more challenging for the RA agent. This setup incorporates the RA agent alongside the multiple NA agents with different language models. We used the RA agent built on Mistral, trained with DG and prompted with LSW, since it achieved the highest win-rate in RQ1. This choice was driven by the limited resources available for training, which required us to focus subsequent experiments on one agent configuration. Each competition consists of 50 games, initialized with a query not used in the training set and a corresponding initial document. Each game spans 30 rounds, which prior work has shown to be sufficient for convergence in LLM-based ranking competitions (Mordo et al., 2025a).

---

[8]Exploring alternative optimization methods, such as GRPO (Guo et al., 2025a), is left for future work.

[9]A win means being ranked the highest for a round.

[10]A subtle consideration arises in the Ho setup. Since the opponents are identical NA agents, their wins are distributed equally among them. This can lead to an extreme case in which the RA agent performs exactly the same as every instance of the NA agent, yet — because of the duplication of opponents — it appears that the RA agent outperforms each of them individually. To account for this effect, we include in Appendix E a dedicated 1-vs-1 competition between the RA agent and a NA agent.

## 5.1 RQ1: EFFECTIVENESS OF THE RA AGENT IN RANKING COMPETITION

To address RQ1, we evaluate the effectiveness of our RA agent in comparison to NA agents in a ranking competition that is conducted over multiple rounds. The evaluation is conducted in the LEMSS simulator for ranking competitions. We trained four lightweight language models: Mistral, Gemma, Llama, and Qwen. We used two distinct data generation methods: SG (Static Generation) and DG (Dynamic Generation); see Section 3 for more details on these generation methods. In SG, the pseudo-relevant document for each query was modified five times. In the DG setup, we first simulated a competition with 450 games (one game per query), each consisting of 30 rounds and five instantiations of NA agents. We used the generated documents as a training dataset. For both generation methods we used a temperature of 0.8 (Yuan et al., 2023). Consistent with prior work (Mordo et al., 2025b; Bardas et al., 2025), we employed PAW and LSW as the prompting strategies, and used the unsupervised E5 ranking function (Wang et al., 2024b) for both data generation and evaluation.

Gemma, Llama, and Qwen were trained only under the DG setup with the LSW prompt following initial evaluation in which we ran a competition with the base (non-RL) versions of all four models. Mistral was the worst-performing model in this initial evaluation, and was therefore selected for a broader configuration analysis, including the SG and the PAW prompt. The motivation for focusing on Mistral was to demonstrate that even if the underlying LLM performs the worst in an initial evaluation, it is still possible to design an RA agent that outperforms NA agents based on other LLMs.

Table 1 presents the win-rate comparisons across different competition configurations. In all cases, the RA agent outperforms the random baseline (20% win-rate). Moreover, in nearly all scenarios, the RA agent significantly outperformed the best NA agent[11]. Notably, the RA agent fine-tuned on Mistral with the LSW prompt achieved the highest win rates under both Ho and He settings (0.75 and 0.6, respectively). Among agents trained with DG, Table 1 shows consistently higher performance in the Ho setting compared to He. This can be attributed to the alignment between the agent's underlying language model and those used by its competitors and for data generation in the Ho case. In contrast, the He setting includes heterogeneous agents based on different underlying LLMs, thereby introducing more diverse documents that challenge our RA agent to adapt its strategy effectively. In Appendix E, we extend our analysis and demonstrate that the performance of the RA agent remains robust with respect to both the number of competitors and the evaluation-time temperature of the LLM.

As shown in Appendix F, the performance of our RA agents is not sensitive to the choice of the DPO hyperparameter $\beta$: across all tested values, RA agents consistently outperform their NA agents counterparts. Appendix G further demonstrates that RA agent's performance improves steadily as the number of queries used for fine-tuning increases, with clear gains even from a relatively small number of preference pairs. Together, these results indicate that RA agents (i) are not sensitive to reasonable $\beta$ choices and (ii) already achieve meaningful performance with modest training data, while continuing to improve as more pairs are provided.

A comparison between SG and DG in Table 1 highlights two distinct aspects of the designing of the RA agent. SG primarily focuses on aligning the agent with the ranker by learning which document variants are preferred, but it does not account for the evolving strategies of other competitors. In contrast, DG explicitly models the dynamic nature of the task by simulating multi-round competitions in which agents continuously adapt their modifications in response to rankings. This distinction is reflected in Table 1, where DG-trained agents consistently outperform their SG counterparts – most notably for Mistral. The RA agent in the setting with the LSW prompt and DG procedure achieves a win-rate of 0.60 under He setting and 0.75 under the Ho setting, compared to 0.29 for SG in He. For the PAW prompt the trends are similar. SG under the He setting achieves a win-rate of 0.29 while the DG achieves 0.36. These results indicate that designing a competitive agent cannot be reduced to the static task of learning the ranker alone; rather, it also requires learning effective strategies against adaptive opponents.

To further contextualize these findings, we additionally explored the document-modification strategies employed by the agents in Appendix I. Our analysis revealed that in the DG setting, greater diversity in ranked lists was observed for the RA agent compared to the NA agent. This effect arises because

---

[11]Except for the case of Llama trained with DG, using the LSW prompt and evaluated under the He setting

Table 1: Comparison of agent performance under heterogeneous (He) and homogeneous (Ho) configurations. We report the win-rate (**WR**) of the RA agent (RL-aligned agent) and the best NA agent (non-aligned agent). '*' marks a statistically significant difference with the win-rate of the best NA agent in the same configuration. The best performance in each configuration is boldfaced.

| LLM | Train Setting | Heterogeneous | | Homogeneous | |
|---|---|---|---|---|---|
| | | RA agent WR | Best NA agent WR | RA agent WR | Best NA agent WR |
| Mistral | SG (PAW) | **0.29**$^*$ | 0.21 | **0.29**$^*$ | 0.2 |
| Mistral | SG (LSW) | **0.29**$^*$ | 0.20 | **0.58**$^*$ | 0.17 |
| Mistral | DG (PAW) | **0.36**$^*$ | 0.20 | **0.71**$^*$ | 0.13 |
| Mistral | DG (LSW) | **0.60**$^*$ | 0.11 | **0.75**$^*$ | 0.10 |
| Gemma | DG (LSW) | **0.34**$^*$ | 0.19 | **0.54**$^*$ | 0.15 |
| Llama | DG (LSW) | 0.24 | 0.24 | **0.59**$^*$ | 0.14 |
| Qwen | DG (LSW) | **0.33**$^*$ | 0.18 | **0.49**$^*$ | 0.16 |

the RA agent makes more substantial document modifications across rounds, leading also to lower similarity between successive documents' versions than in the NA agents. In contrast, the SG setting yields more homogeneous documents and similar modification patterns for both RA agent and NA agent. Consistent with prior work Mordo et al. (2025a), both agent types eventually converge toward stable documents.

We also analyzed how competition affects both the win-rate and the relevance judgments of the RA agent and NA agents (Appendices I and K). Relevance annotations and win-rate analyses show that the stronger alignment of the RA agent with the ranker provides a clear advantage at the start of the competition: in round 1, the RA agent produces documents of significantly higher relevance and achieves higher win-rates than the NA agent. By round 30, however, this advantage reduced as NA agents improve through competition — an instance of the herding effect Raifer et al. (2017), where all agents gravitate toward similar highly relevant documents. Notably, in the SG setting, the advantage of the RA agent relative to the NA agent is substantially reduced compared to DG.

We extend the study to competitions involving multiple RA agents in Appendix J. When multiple RA agents compete, their presence increases the inter-document similarity in ranked lists but does not significantly affect overall ranking performance, suggesting that diverse adaptation strategies primarily emerge in multiple-agents settings.

Finally, in Appendix H, we further evaluate our agent in the single-round setting of Bardas et al. (2025). The results show that our RA agent consistently outperforms the NA agents in both ranking promotion and content faithfulness. Together with the repeated-competition evaluation, these findings demonstrate that the advantages of our RA agent extend across several competitive settings.

**Faithfulness Analysis**   We analyzed the faithfulness of agents-modified documents to their original versions over 30 competition rounds, averaging scores across queries. We evaluated the RA agent and the NA agent in the configurations prompted with LSW and instantiated with the Mistral language model. The configurations included DG under both He and Ho, and SG under He. The comparison of the faithfulness between the RA agent and the NA agent is shown in Figure 2. In the early rounds (Rounds 1–4), both agents in all settings maintain relatively high faithfulness, with scores above 0.5[12]. In addition, across most rounds, the RA agent consistently achieves higher faithfulness than the NA agent. Toward later rounds, both agents exhibit converging faithfulness trends, reflecting limited further document modifications, a phenomenon consistent with observations in prior work (Mordo et al., 2025a).

Overall, our results suggest that the RA agent not only outperforms the NA agent in win-rate, but also better preserves the faithfulness to the original document throughout the competition.

---

[12]I.e., more than 50% of the sentences are entailed by the initial document (Gekhman et al., 2023).

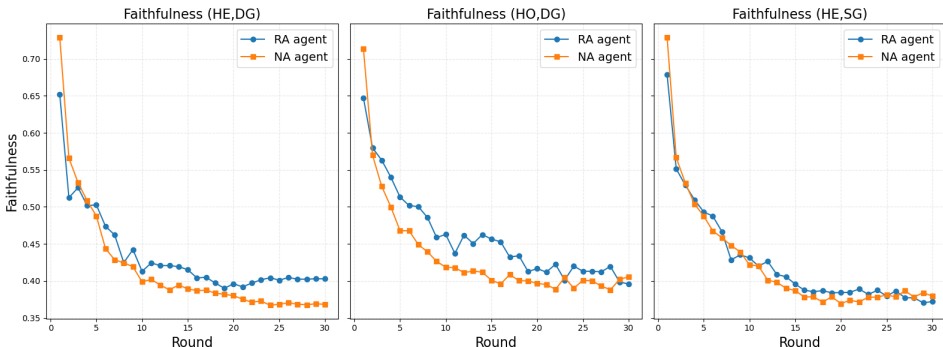

Figure 2: The faithfulness score of the RA agent and the NA agent for the He and DG (left), Ho and DG (middle), and He and SG (right) settings.

Table 2: Comparison of the win-rate (**WR**) in the He competitions with Mistral 8B agents trained with DG and prompted with LSW under different ranking functions used for training and evaluation. We report the win-rate of the RA agent (RL-aligned agent) and the best NA agent (non-aligned agent). '*' marks a statistically significant difference with the win-rate of the best NA agent.

| LLM | Train Setting | Trained Ranker | Tested Ranker | RA agent WR | Best NA agent WR |
|---|---|---|---|---|---|
| Mistral | DG (LSW) | E5-unsupervised | E5-supervised | **0.27***  | 0.20 |
| | | | Contriever | **0.28** | 0.25 |
| | | | Okapi | **0.29***  | 0.21 |
| | | Contriever | E5-supervised | **0.44***  | 0.17 |
| | | | E5-unsupervised | **0.50***  | 0.15 |
| | | | Okapi | **0.58***  | 0.12 |

## 5.2 RQ2: TRANSFER LEARNING ACROSS RANKING FUNCTIONS

In RQ2, we study the extent to which the performance of the RA agent generalizes across ranking functions, specifically when there is a mismatch between the ranker used during training and the one used during evaluation. This setting reflects realistic deployment scenarios, where the true ranking function may differ from the one used during development or may even change over time. Hence, robustness to ranker shifts is a key requirement for practical applicability. We focus on the best RA agent from RQ1: the Mistral language model, trained using the DG procedure and the LSW prompt.

We trained the agent using two different ranking functions: E5-unsupervised (Wang et al., 2024b), and Contriever (Izacard et al., 2022). Evaluation was conducted under the He competition setting, using each of the aforementioned rankers as well as two additional rankers: (1) a supervised variant of E5 (Wang et al., 2024b), to study the impact of supervision in the ranking function, and (2) Okapi BM25 (Robertson et al., 1993).

Table 2 presents the win-rate results of the RA agent and the NA agent across the various combinations of training and evaluation ranking functions. In almost all relevant comparisons, the RA agent significantly outperformed the best NA agent in the competition, attesting to its ability to transfer effectively across rankers, even when they were not used for training. Interestingly, the results reveal that transfer learning across ranking functions is asymmetric. For instance, when the RA agent is trained using the E5-unsupervised ranker and evaluated on Contriever, it achieves a win-rate of 0.28. In contrast, when trained with Contriever and evaluated using E5-unsupervised, the win-rate increases to 0.50. This asymmetry suggests that certain rankers may induce more generalizable training signals than others. All in all, these findings highlight both the robustness and the directional sensitivity of transfer learning of our RA agents in repeated ranking games.

## 6 CONCLUSION

We introduced an RL-aligned (RA) agent for competitive search, where LLMs act as publishers in repeated ranking games. Our extensive experiments show that our agent consistently outperforms non-aligned (NA) agents, demonstrating the effectiveness of RL in this strategic retrieval setting. For future work, we intend to pursue several directions. First, devising alternative optimization strategies and loss formulations specifically tailored to ranking-based alignment is a promising avenue for improving agent performance. Second, we plan to design RL-based strategies that explicitly encourage higher levels of faithfulness, with the goal of balancing ranking effectiveness and faithfulness to the original document. Finally, we aim to explore online agents that can learn and adapt during the ranking competition itself, rather than being trained solely before test time.

**Ethics Statement** This research does not involve human subjects, personal data, or sensitive information, and therefore does not raise privacy, security, or IRB-related concerns. All datasets used are publicly available (e.g., MS MARCO, TREC) or synthetically generated by large language models, and no copyrighted or proprietary data was included. Our experiments focus on ranking competitions in a controlled simulation framework and do not involve deployment in real-world systems. While our work introduces reinforcement learning strategies to optimize LLM-based agents in competitive search, we acknowledge that ranking manipulation and strategic content generation may raise concerns if misused. To mitigate such risks, we restrict our study to academic evaluation settings and will release in the camera-ready version code and data solely for reproducibility and further research in information retrieval and responsible AI. We also note that both large language models and ranking functions may reflect societal biases present in their training data. Although addressing bias and fairness is not the primary focus of this work, we encourage future studies to examine how such factors interact with strategic content generation in competitive search.

**Reproducibility Statement** We provide a detailed description of our algorithms in Section 3, with additional technical details in Appendices B and C.2. Hyper-parameters for dataset generation and agent training are reported in Appendix C. All evaluation measures are well-defined (see Appendix D) to facilitate replication. The datasets we used for evaluation, as well as the code for analysis, data generation, and agent design will be released with the camera-ready version to ensure full reproducibility.

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

## A    PRELIMINARIES

**Reinforcement Learning (RL)**    A Markov Decision Process (MDP) is defined as a tuple $(\mathcal{S}, \mathcal{A}, P, r, T, \gamma)$, where $\mathcal{S}$ is the state space, $\mathcal{A}$ is the action space, $P : \mathcal{S} \times \mathcal{A} \rightarrow \Delta(\mathcal{S})$ is the transition probability function, $r : \mathcal{S} \times \mathcal{A} \rightarrow \mathbb{R}$ is the reward function, $T$ is the episode horizon, and $\gamma \in [0, 1]$ is the discount factor. An agent interacts with the environment through a stationary stochastic policy $\pi : \mathcal{S} \rightarrow \Delta(\mathcal{A})$ that maps each state to a distribution over actions. The value of a policy $\pi$ at a state $s$ is the expected discounted return, defined as

$$V^{\pi}(s) = \mathbb{E}_{P,\pi}\left[\sum_{t=0}^{T-1} \gamma^t r(s_t, a_t) \,\middle|\, s_0 = s\right].$$

The objective in reinforcement learning is to find an optimal policy $\pi^*$ that maximizes the expected value over an initial state distribution $\nu_0$, that is,

$$\pi^* \in \arg\max_{\pi} \mathbb{E}_{s_0 \sim \nu_0}\left[V^{\pi}(s_0)\right].$$

**Large Language Models (LLMs)**    A large language model (LLM) $\mathcal{L} : \mathcal{S} \mapsto \Delta_{\mathcal{S}}$ maps sequences of tokens to probability distributions over future sequences. These models are typically implemented using Transformer architectures (Vaswani et al., 2017), and are trained to predict the next token $x_t$ in a sequence, given the preceding tokens $(x_1, x_2, \ldots, x_{t-1})$, by minimizing the cross-entropy loss. Pre-trained LLMs vary significantly in size and capabilities, with larger models often exhibiting stronger reasoning, generalization, and generation performance. For example, the LLaMA 2 series (Touvron et al., 2023) includes models with 7B, 13B, and 70B parameters.

**Direct Preference Optimization (DPO)**    To align LLMs with human preferences, Direct Preference Optimization (DPO; Rafailov et al., 2024) provides a direct alternative to reinforcement learning methods such as PPO. DPO is usually trained on a dataset of human preferences in the form of tuples $(x, y_w, y_l)$, where $x$ is a prompt, $y_w$ is a preferred response, and $y_l$ is a less preferred one. Instead of using explicit reward modeling or rollout trajectories, DPO optimizes a contrastive loss that directly encourages the policy $\pi_\theta$ to assign higher likelihood to the preferred response relative to a reference policy $\pi_{\text{ref}}$. The DPO objective is defined as:

$$\mathcal{L}_{\text{DPO}}(\pi_\theta; \pi_{\text{ref}}) = -\mathbb{E}_{(x,y_w,y_l)\sim\mathcal{D}}\left[\log \sigma\left(\beta \log \frac{\pi_\theta(y_w \mid x)}{\pi_{\text{ref}}(y_w \mid x)} - \beta \log \frac{\pi_\theta(y_l \mid x)}{\pi_{\text{ref}}(y_l \mid x)}\right)\right],$$

where $\sigma$ is the sigmoid function, $\beta > 0$ is a temperature parameter controlling the sharpness of the preference, $\pi_{\text{ref}}$ is typically set to the pre-trained base model and $D$ is a distribution over datapoints. This formulation introduces implicit regularization by comparing against the reference model and enables stable and efficient fine-tuning of LLMs using preference data, without requiring reinforcement learning rollouts or reward modeling.

**A Schematic Figure of a Ranking Game**

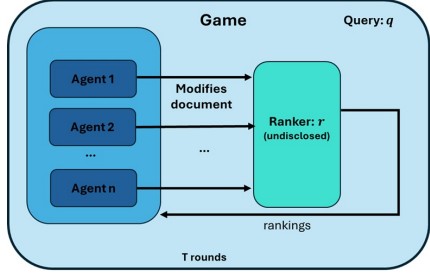

Figure 3: Illustration of a single game within a ranking competition. Each competition consists of multiple games. Each game is assigned with a query and composed of multiple rounds of agents' interaction. In each round, agents modify their documents and receive the rankings of each document.

# B  DATA GENERATION

## B.1  STATIC GENERATION (SG)

In this approach, we generate multiple relevant documents per query using an LLM prior to any optimization phase. We then collect the top-ranked and lowest-ranked documents (per query) to construct training triplets in the form (prompt, top-ranked document, lowest-ranked document).

We first prompt the LLM to generate a single relevant document for a given query, referred to as the *pseudo-relevant document*. To ensure neutrality with respect to the ranking competition, we employ the instructional (system) prompt proposed by Bardas et al. (2025), omitting any mention of competitive context. We then apply the Best-and-Worst-of-N (BWoN) sampling method, adapted from the Best-of-N strategy (Beirami et al., 2025). Given the pseudo-relevant document, we prompt the LLM $N$ times to generate $N$ *modified documents* of the pseudo-relevant document. These documents are ranked using a ranking function. We collect the top-ranked and lowest-ranked documents from this set to construct training triplets in the form (prompt, top-ranked document, lowest-ranked document). This procedure is repeated for each query, yielding a dataset of preference pairs for downstream training. This method assumes access to a ranking function but no additional information about the competition dynamics or other participating agents. The prompts used to generate both the pseudo-relevant and the modified documents are presented in Figures 4 and 5, respectively.

```
PROMPT = "Create a document to achieve high ranking for the candidate query, aiming for the
    highest rank.
Target a document length of around 147 words, not exceeding 150 words.
Generate only a document, without additional comments or titles.

Input:

 - Candidate Query: {}

The document:"
```

Figure 4: The prompt for generating the pseudo-relevant document.

```
PROMPT = "You are participating in a search engine optimization (SEO) process.
Edit the candidate document to improve its search engine ranking for the candidate query,
    aiming for the highest rank.
Focus on editing the most impactful sentences to enhance ranking potential.
Target an edited document length of around 147 words, not exceeding 150 words.
Ensure the edited document is very similar to the candidate document. Generate only the edited
    document, without additional comments or titles.

Input:

 - Candidate Query: {}

 - Candidate Document: {}

Edited Document: "
```

Figure 5: The prompt for generating the modified documents with no past rankings feedback.

## B.2  DYNAMIC GENERATION (DG)

While the static approach ignores the documents and rankings of other agents, the DG method explicitly models the dynamics of the competition. It does so by incorporating the documents and rankings of competing agents through simulations of repeated ranking games within the LEMSS environment (Mordo et al., 2025a). We instantiate multiple copies of the LLM to simulate an $N$-player competition. The initial document in each training episode is generated by the LLM, following the same procedure as generating the pseudo-relevant document in SG. This choice, inspired by Zhou

et al. (2024), mitigates the off-policy distribution mismatch that can occur when the agent encounters states it has never seen during training; by ensuring the RA agent learns from inputs representative of its training environment, we reduce instability and improve learning efficiency. In contrast, for evaluation we used initial documents drawn from a fixed dataset, ensuring that all agents received the same initial document for each query. This setup guarantees a common starting point and enables a fair comparison of strategies, following the standard approach adopted in prior work on competitive search (Raifer et al., 2017; Mordo et al., 2025b). For each query and round selected from a set of rounds, we log the prompt presented to our agent, and extract the documents submitted by the highest- and lowest-ranked agents.

The resulting preference dataset consists of prompt-document triplets where the top and lowest ranked documents reflect actual competitive outcomes based on the simulated ranking environment. These data generation methods can be interpreted along the level of the agent's awareness of its downstream task and environment. As more contextual information becomes available, such as the identity or number of competing agents, the generated data increasingly approximates the true target distribution encountered during actual ranking competitions. Importantly, each method involves an inherent trade-off between exploration and sample efficiency: increasing the number of samples generated by the LLM can enhance exploratory coverage of the document space, thereby potentially improving the diversity of documents' scores of the resulting training data with respect to the ranking function. A systematic investigation of this trade-off is an important direction for future work.

### B.3 PARAMETERS

Document generation was performed using a temperature of 0.8 to control sampling diversity (Yuan et al., 2023). For both generation methods, we adopted the LSW and PAW prompts (Bardas et al., 2025). In the SG method, we generated five modified documents per query and extracted training triplets consisting of the prompt, the top-ranked document, and the lowest-ranked document, based on a predefined ranking function. In the DG method, each simulated game involved five agents and lasted for 30 rounds, following Mordo et al. (2025a). To match the dataset size of SG, we selected only one round per query: round 3 for LSW and round 4 for PAW, as these are the first rounds with full ranking history required for the respective prompts.

## C  HYPER-PARAMETERS

We report the hyper-parameters used in all generation and training phases.

### C.1  GENERATION SETTINGS

For all generative agents, we used the following decoding parameters during document generation:

- **Temperature:** 0.8
- **Top-p (nucleus sampling):** 1.0          *(as recommended in the TRL library)*
- **Top-k:** 0          *(disables top-k filtering; used with top-p)*

### C.2  GENERAL TRAINING SETTINGS

We trained each LLM with two distinct datasets: SG and DG. Each LLM was fine-tuned on the 20 last transformer layers using the Transformer Reinforcement Learning (TRL) library (von Werra et al., 2020) and the DeepSpeed optimization framework (Rasley et al., 2020). Preliminary experiments indicated that fine-tuning fewer layers resulted in suboptimal performance, whereas deeper fine-tuning led to consistent improvements. We therefore selected 20 layers as a practical trade-off, given available resources. Due to computational constraints, some of training hyper-parameters were manually chosen with default values rather than tuned through extensive optimization. Our primary goal in this work is to establish and validate the alignment framework, rather than to exhaustively optimize agent performance. Nevertheless, as demonstrated in RQ1, even without hyper-parameter tuning, we successfully designed an RA agent that outperforms the NA agent. We used the following optimization configuration:

- **Batch size:** 2
- **Gradient accumulation steps:** 4
- **Number of epochs:** 4
- **Learning rate:** $1 \times 10^{-6}$
- **Number of trainable transformer layers:** 20
- **Loss:** WPO (weighted DPO variant; Zhou et al., 2024)
- **DPO/WPO beta:** 0.1

We used the Adam optimizer with the following configuration:

- **Beta 1:** 0.9
- **Beta 2:** 0.99
- **Weight decay:** 0.01

We emphasize that training RA agents requires simulating a competition consisting of at least $k + 1$ rounds, where $k$ denotes the history depth provided to the agent as context input during the competition. Consequently, for the LSW and PAW prompts, three and four rounds are required, respectively. To achieve this, one can employ a single LLM shared across two A100 40GB GPUs with a batch size of 16, resulting in 85 and 113 inferences per competitor, respectively; the number of inferences are the result of: datasetSize * historyDepth / batchSize.

## D EVALUATION MEASURES

**Scaled Promotion**    To quantify how effectively a document modification improves ranking within a single round, we use the *Scaled Promotion* metric. It measures the normalized improvement (or demotion) in rank between consecutive rounds:

$$\text{Scaled Promotion}_t(d) = \frac{\text{Rank}_t(d) - \text{Rank}_{t+1}(d)}{\max\left(\text{Rank}_t(d) - 1,\ N - \text{Rank}_t(d)\right)} \tag{1}$$

where $\text{Rank}_t(d)$ is the rank of document $d$ in round $t$, $\text{Rank}_{t+1}(d)$ is its rank in the following round, and $N$ is the number of competing documents. The denominator represents the maximum achievable promotion (if the document is not ranked first) or demotion (if it is not ranked last). A higher score indicates a stronger relative promotion, normalized by what is theoretically possible.

**OrigFaith (faithfulness to the original document)**    Given an original (initial) document $d_{\text{orig}}$ and a modified document $d_{\text{mod}} = \{s_1, \ldots, s_m\}$ with $m$ sentences, we first compute the *Raw Faithfulness* score using an NLI-based model (TrueTeacher, TT) (Gekhman et al., 2023):

$$\text{RawFaith}(d_{\text{mod}}, d_{\text{orig}}) = \frac{1}{m} \sum_{i=1}^{m} \mathbb{1}\big\{\text{TT}(s_i, d_{\text{orig}}) \geq 0.5\big\}. \tag{2}$$

where $\text{TT}(s_i, d_{\text{orig}}) \in [0, 1]$ is the entailment probability between the modified sentence $s_i$ and the original document $d_{\text{orig}}$, and $0.5$ is a predefined entailment threshold chosen according to Gekhman et al. (2023).

To account for varying document lengths and ensure comparability across instances, we normalize the RawFaith score:

$$\text{OrigFaith}(d_{\text{mod}}, d_{\text{orig}}) = \frac{\text{RawFaith}(d_{\text{mod}}, d_{\text{orig}})}{\text{RawFaith}(d_{\text{orig}}, d_{\text{orig}})}, \tag{3}$$

This yields a normalized faithfulness score in $[0, 1]$ that reflects how well the modified document preserves the faithfulness to the original document.

**Win-rate**    This metric measures how frequently an agent achieves the top rank across rounds, averaged over all queries (games). It is defined as:

$$\text{Win Rate} = \frac{1}{|Q|} \sum_{q=1}^{|Q|} \frac{W_q}{R_q} \tag{4}$$

where $|Q|$ is the number of queries in the evaluation set, $W_q$ is the number of rounds in which the agent ranked first for query $q$, and $R_q$ is the total number of rounds played for query $q$. This metric captures the agent's ability to consistently produce top-ranked outputs relative to its competitors. We report the win-rate of the RA agent and compare it against two baselines: (i) a random baseline, equal to $\frac{1}{\#\text{players}}$, and (ii) the NA agent with the best performance with respect to the win-rate.

Table 3: Comparison of performances in Ho competitions with Mistral 8B agents trained with DG and prompted with LSW under different number of NA agents compete the RA agent. We report the win-rate of the RA agent and the best NA agent. '*' marks a statistically significant difference with the win-rate of the best NA agent. The best performance in each configuration is boldfaced.

| LLM | Train Setting | Temp. | # NA agents | RA agent WR | Best NA agent WR |
|---|---|---|---|---|---|
| Mistral | DG (LSW) | 0.5 | 1 | **0.72**$^*$ | 0.28 |
| | | | 4 | **0.65**$^*$ | 0.11 |
| | | | 7 | **0.65**$^*$ | 0.06 |
| | | 1 | 1 | **0.74**$^*$ | 0.26 |
| | | | 4 | **0.60**$^*$ | 0.11 |
| | | | 7 | **0.58**$^*$ | 0.07 |

Table 4: Comparison of performances in He competitions with Mistral 8B agents trained with DG and prompted with LSW under temperatures of the LLM at evaluation time. We report the win-rate of the RA agent and the best NA agent. '*' marks a statistically significant difference with the win-rate of the best NA agent. The best performance in each configuration is boldfaced.

| LLM | Train Setting | Temp. | RA agent WR | Best NA agent WR |
|---|---|---|---|---|
| Mistral | DG (LSW) | 0.5 | **0.60**$^*$ | 0.11 |
| | | 0.8 | **0.58**$^*$ | 0.11 |
| | | 1 | **0.62**$^*$ | 0.11 |
| | | 1.5 | **0.62**$^*$ | 0.15 |
| | | 2 | **0.58**$^*$ | 0.11 |

# E  ROBUSTNESS OF THE RA AGENT PERFORMANCE

We evaluate the robustness of the RA agent with respect to two key competition parameters: (1) the number of competing agents, and (2) the sampling temperature of the agent's LLM at evaluation. Studying these aspects is crucial for understanding whether a trained agent remains effective when deployed under varying and potentially unpredictable conditions. For example, in practical environments, the number of competitors and the behavior of LLM-based agents (e.g., due to randomness introduced by sampling) may fluctuate significantly. Thus, an agent's resilience to such changes is an important factor in its practical utility. We focus on the best-performing agent from RQ1 (See Section 5.1.): a Mistral-based model trained using the DG procedure with LSW prompting. For both training and evaluation, we use the E5-unsupervised ranker (Wang et al., 2024b), which demonstrated superior performance in past work over other ranking functions, and has also been adopted in prior work on competitive search (Mordo et al., 2025b; Bardas et al., 2025).

Table 3 reports the win rates of our agent in competitions with 1, 4, and 7 competitors. The results are presented for the Ho setting, in which each competitor is a duplication instance of the same NA agent. We did not consider the He setting in order to isolate the effect of the number of agents from potential confounding factors related to the choice of language model. We evaluate the agent at two sampling temperatures: 0.5 and 1.0. Temperature 0.0, used in previous RQs, is omitted here as it prevents exploration of stochastic behavior in competitive settings. Across all configurations, the RA agent consistently outperforms the best NA agent. As expected, the win-rate decreases with the number of competitors due to increased competition, but remains significantly above the random baseline.

Table 4 presents the results of a broader temperature sweep, evaluating the agent at temperatures 0.0, 0.5, 0.8 (matching the temperature used during data generation), 1.0, 1.5, and 2.0. We fixed the number of competitors as five, under the He setting. In all tested temperatures, our agent maintains a win-rate in the range $[0.58, 0.62]$, significantly outperforming all competitors across the board. These findings demonstrate that the RA agent is robust to variation in both the number of competitors and the temperature at the evaluation phase.

Table 5: Comparison of performances in Ho competitions with Mistral 8B agents trained with DG and prompted with LSW under different values of $\beta$. We report the win-rate of the RA agent and the best NA agent. '*' marks a statistically significant difference with the win-rate of the best NA agent. The best performance is boldfaced.

| LLM | Train Setting | $\beta$ | RA agent WR | Best NA agent WR |
|---|---|---|---|---|
| | | 0.05 | **0.49**$^*$ | 0.17 |
| | | 0.1 (original) | **0.75**$^*$ | 0.10 |
| Mistral | DG (LSW) | 0.2 | **0.69**$^*$ | 0.11 |
| | | 0.3 | **0.67**$^*$ | 0.12 |

Table 6: Comparison of performances in Ho competitions with Mistral 8B agents trained with DG and prompted with LSW under different number of queries in the training set. We report the win-rate of the RA agent and the best NA agent. '*' marks a statistically significant difference with the win-rate of the best NA agent. The best performance is boldfaced.

| LLM | Train Setting | Number of Queries | RA agent WR | Best NA agent WR |
|---|---|---|---|---|
| | | 50 | **0.51**$^*$ | 0.24 |
| | | 100 | **0.67**$^*$ | 0.16 |
| | | 150 | **0.56**$^*$ | 0.22 |
| | | 200 | **0.73**$^*$ | 0.1 |
| Mistral | DG (LSW) | 250 | **0.67**$^*$ | 0.16 |
| | | 300 | **0.72**$^*$ | 0.12 |
| | | 350 | **0.74**$^*$ | 0.11 |
| | | 400 | **0.82**$^*$ | 0.06 |
| | | 450 | **0.75**$^*$ | 0.10 |

## F  SENSITIVITY TO $\beta$

Table 5 reports the sensitivity of the DPO alignment stage to the choice of the $\beta$ parameter. The Mistral RA agents were trained using DG configuration and a Ho evaluation setting with four competing NA agents, we evaluated RA agents across a range of $\beta$ values. The RA agent consistently outperformed the NA agents, with win rates spanning 0.50 to 0.75, and the differences were statistically significant for all tested $\beta$ values. This indicates that the RA agent performance is not overly sensitive to $\beta$ and that the alignment benefits persist across a broad and reasonable range of hyperparameter choices.

## G  SAMPLE COMPLEXITY

To quantify how the amount of queries in the training data affect the RA agent performance, we conducted a sample complexity study using the Mistral LLM trained with DG (Dynamic Generation) datasets, prompted with LSW. We trained RA agents with datasets containing 50 to 450 queries along with document pairs, increasing in steps of 50, and evaluated each agent in a Ho competition setting with four NA agent Mistral opponents; the results are reported in Table 6 and Figure 6. The RA agent's win-rate increases consistently as more training samples are provided, ranging from 0.51 to 0.82, and begins to exhibit signs of convergence at larger sample sizes. These results show that RA agents can achieve meaningful performance gains over NA agents with a moderate number of training pairs, and that improvements continue — albeit with diminishing returns — as more pairs are added.

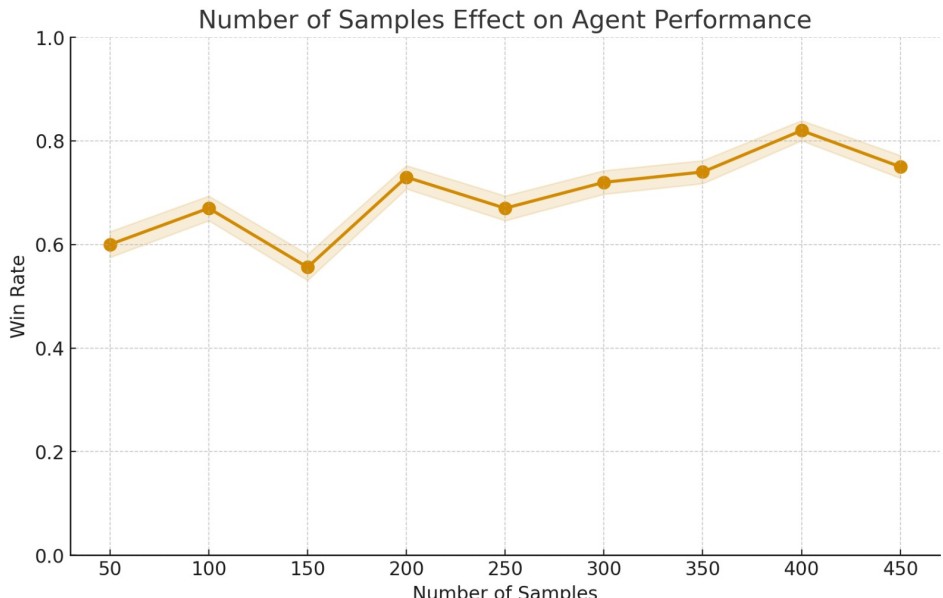

Figure 6: The win-rate of Mistral RA agents trained with DG datasets and prompted with LSW as a function of the number of queries in the training data. The agents were evaluated in a Ho competition setting with four NA agent Mistral opponents. The confidence interval was computed using Bernoulli variables.

## H  EFFECTIVENESS OF THE RA AGENT IN SINGLE-ROUND OFFLINE EVALUATION

We evaluate the RA agent and compare its performance to that of NA agents in the single-round setting introduced by Bardas et al. (2025). In this setting, each agent modifies documents in the context of an existing competition previously conducted between students (Mordo et al., 2025b); the students were rewarded to improve their rankings. We consider an RA agent with Mistral 8B language model with the LSW prompt, trained using the DG procedure with the E5-unsupervised ranking function; the same ranker was used in ranking competitions with human participants (Mordo et al., 2025b). The NA agent used also the LSW prompt.

We report two evaluation measures: scaled promotion and faithfulness. The scaled promotion metric is used to quantify ranking properties, computed per player and her document for a query. Specifically, it measures the change in a document's rank between consecutive rounds, defined as the number of positions by which the document is promoted (or demoted), normalized by the maximum potential promotion (or demotion) given the document's position. The values for the students are averaged over them and the queries, while the values for an agent is averaged over queries. The faithfulness[13] captures whether the modifications preserve the factual consistency of the original document; it is measured using the NLI-based approach proposed by Gekhman et al. (2023). Formal definitions are presented in Appendix D.

Table 7 shows that among the agents with Mistral 8B, the RA agent achieves scaled promotion that is higher than that of the student[14] participants and the NA agent. Additionally, the RA agent also demonstrates higher faithfulness to the original document than the NA agent. However, both the RL and NA agents exhibit faithfulness scores lower than those of human participants. This indicates that while LLM-based agents are effective at strategic promotion, they may struggle to preserve content faithfulness relative to human baselines.

---

[13]See Section 4.4.

[14]Note that student scores vary across rows as they depend on the agent under evaluation.

Table 7: Performance comparison of Mistral 8B RA agent and NA agent using the LSW prompt. The RA agent was trained with the DG procedure. The Table presents scaled promotion and faithfulness scores from a single-round offline evaluation conducted on an existing ranking competition following the setup of Bardas et al. (2025).

| LLM | Scaled Promotion | | Faithfulness | |
|---|---|---|---|---|
| | Students | The Agent | Students | The Agent |
| Mistral 8B + RL | 0.089 | 0.266 | 0.788 | 0.408 |
| Mistral 8B | 0.328 | $-0.363$ | 0.788 | 0.350 |

Table 8: Performance comparison of NA agents with larger (than 8B) language models. The table presents scaled promotion and faithfulness scores from a single-round offline evaluation conducted on an existing ranking competition following the setup of Bardas et al. (2025).

| LLM | Scaled Promotion | | Faithfulness | |
|---|---|---|---|---|
| | Students | The Agent | Students | The Agent |
| Llama 70B | 0.100 | 0.270 | 0.788 | 0.785 |
| Qwen2.5 32B | 0.226 | $-0.119$ | 0.788 | 0.666 |
| Gemma2 27B | 0.086 | 0.296 | 0.788 | 0.936 |

Additionally, the scaled promotion and faithfulness scores of the RL and NA agents are lower than those reported by Bardas et al. (2025) for gpt-4-based agents. This performance gap is attributed to the use of language models with 8B parameters in our experiments, a significantly smaller model compared to gpt-4. To support this claim, Table 8 includes results for NA agents with larger language models: Llama 70B, Qwen2.5 32B and Gemma2 27B which indeed outperform both the Mistral 8B variants in both scaled promotion and faithfulness. In future research we intend to explore training methods to optimize not only the ranking promotion but also the faithfulness to the initial document.

We note that our experiments focused on lightweight models, where resource constraints allowed systematic study across multiple rankers and learning settings. Within this scope, we consistently observed that RA agents are more faithful than NA agents, even though our RLRF method is not directly optimized for faithfulness — an interesting emergent result. A full comparison between lightweight and large models would provide additional insight. We included above results for larger NA agents as inspiration, showing that faithfulness tends to increase with model scale. We hypothesize that larger RA agents would exhibit the same trend, as larger LLMs typically follow the instructional prompts given to the LLM to a larger extent (In the prompt to the LLMs we ask all the agents to "...Ensure the edited document is very similar to the candidate document...". However, due to computational limits, we could not currently validate this experimentally.

# I  ANALYSIS OF STRATEGIES

To complement the win-rate results (See Section 5.1), we analyze the underlying strategies that the RA agent and NA agent employ when modifying their documents over time. Our focus is on the settings with Mistral from RQ1: (i) competitions with the LSW prompt under DG, evaluated in both the Ho and He settings, and (ii) the SG generation method under the He setting.

We adopt several measures introduced by Mordo et al. (2025a), chosen to capture both player-level and ranked-list-level dynamics. First, we measured the *diversity* of documents by computing the minimum inter-document similarity within a ranked list across rounds. This measure reflects how varied the documents remain throughout the competition. Second, we evaluated the *convergence* of a competition at the player-level. We measured the similarity between documents produced by the same agent between consecutive rounds. This indicates the extent to which agents continue modifying their documents as the competition progresses, and whether their strategies stabilize over time. Third, we track the *scores* assigned by the ranking function to the documents of both the RA agent and the NA agents. For the NA agents, we arbitrarily selected one representative per competition.

To compute similarity measures, we use S-BERT (Reimers & Gurevych, 2019) as the encoder for document representations and apply cosine similarity to their embeddings. The results as a function of the round are presented in Figures 7 and 8.

Figure 7 shows that the minimum inter-document similarity is consistently higher under the SG setup than under DG. This is expected, since in the static case the agent was trained on a self-generated dataset independent of competitive dynamics, which tends to reduce variation and increase homogeneity across the ranked list. In contrast, the dynamic setup relies on preference data derived from competitions between Mistral clones. This training process exposes the agent to a broader range of document modifications, ultimately fostering greater diversity in the ranked lists.

Figure 8a examines the similarity of each agent's consecutive documents. The RA agent trained with DG (in both Ho and He settings) display the lowest similarity between rounds, indicating that they adapt their documents more substantially across iterations. NA agents, by contrast, exhibit more conservative and homogeneous modifications. In all settings, the RA agent and NA agents demonstrate a tendency to converge toward stable strategies. This convergence is consistent with the herding effect observed in ranking competitions between LLMs (Mordo et al., 2025a), where agents gradually reduce exploration and adopt increasingly similar behaviors.

Finally, Figure 8b plots the ranking function's score assigned to documents over rounds. Across all settings, the RA agent achieves higher scores than the NA agent, reflecting the alignment induced by RLRF training. Notably, NA agents start with relatively low scores but quickly improve during the first few rounds before stabilizing at a plateau. The RA agent, however, is already aligned to the ranking function at the outset, and thus shows smaller relative gains during the competition.

We now turn to analyze the win-rates with respect to the first and last rounds. The first round reflects the initial alignment of the RA agent, while the last round (round 30) captures the dynamics that unfold during the competition. Table 9 reports the win rates of the RA agent and the NA agent in the first and last rounds. In round 1, the RA agent consistently and significantly outperformed the NA agent across all three settings, demonstrating the effectiveness of its alignment procedure. By round 30, the NA agent had improved its win rate in two of the three settings, yet the RA agent still maintained a clear advantage. This improvement of the NA agent is consistent with the herding effect, whereby agents converge toward similar strategies over repeated rounds. Overall, the results show that the alignment process benefits the RA agent in two ways: it enhances its alignment with the ranking function and strengthens its ability to compete against opponents during the ranking competition.

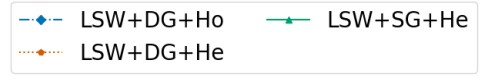
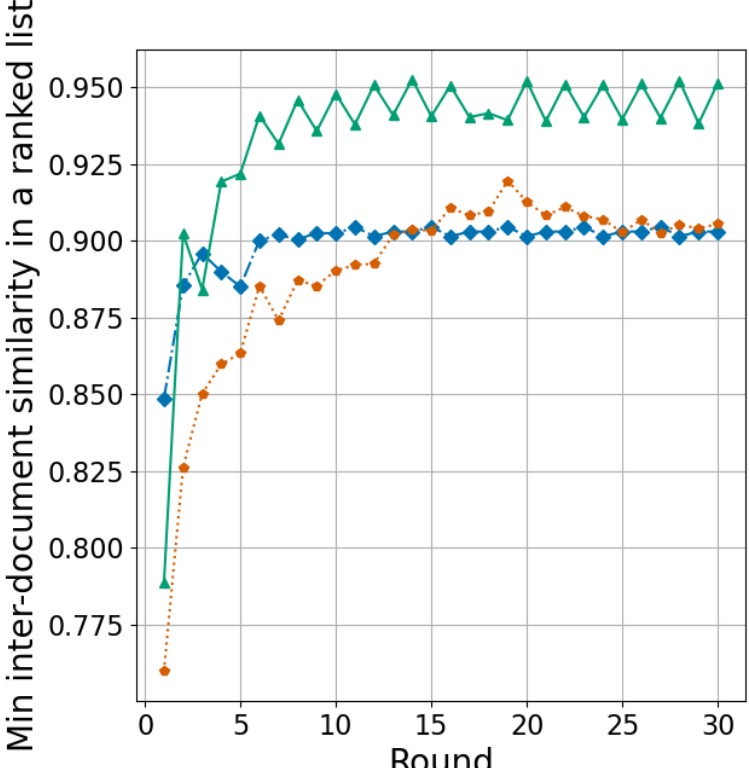

Figure 7: Comparison of the average minimum inter-document similarity in a ranked list across rounds, for the settings in RQ1: (i) DG with the LSW prompt under the Ho setting, (ii) DG with the LSW prompt under the He setting, and (iii) SG under the He setting.

Table 9: Win-rate (WR) of the RA agent and the NA agent at the first and last rounds (1,30) across the configurations and agents. $r$ denotes statistical significance difference between rounds (01 vs. 30) for the same player and setting. $p$ denotes statistical significance difference between the NA and RA agents at the same round and setting.

| Configuration | NA agent | | RA agent | |
| --- | --- | --- | --- | --- |
| | Round 01 | Round 30 | Round 01 | Round 30 |
| LSW+SG+He | 0.12 | 0.20 | $0.50^{p,r}$ | 0.22 |
| LSW+DG+Ho | 0.02 | 0.14 | $0.88^{p}$ | $0.72^{p}$ |
| LSW+DG+He | 0.06 | 0.06 | $0.68^{p}$ | $0.66^{p}$ |

## J    MULTIPLE RA AGENTS

We study the effect of the participation of multiple RA agents in ranking competitions. We focus on RA agents based on Mistral language model, trained using DG and prompted with LSW. All competitions involve five agents with Mistral language models (RA agent and NA agents). In contrast to RQ1, where we used the same LLM hyper-parameters as Bardas et al. (2025), here we set the temperature to 1 (instead of 0) to increase the dynamics of document generation. In each setting, we increment the number of RA agents by one while decreasing the number of NA agents accordingly. We report the same measures as in Appendix I.

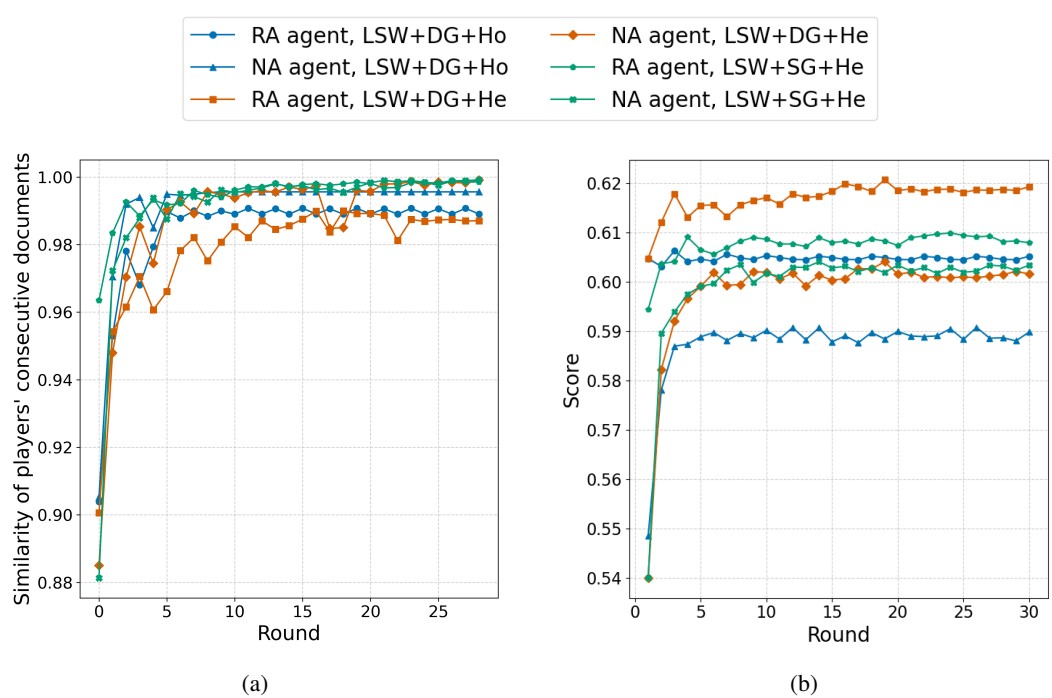

(a)                                                          (b)

Figure 8: Comparison of the RL-aligned agent (**RA agent**) and non-aligned agents (NA agents) under the RQ1 settings: (i) DG with the LSW prompt under the Ho setting, (ii) DG with the LSW prompt under the He setting, and (iii) SG under the He setting. We evaluate the following measures: (a) the average similarity (over queries) between consecutive rounds $(i, i + 1)$, and (b) the average ranking score over rounds (averaged over queries).

Figure 9 shows the minimum inter-document similarity in ranked lists across rounds. In all settings, the similarity in a ranked list increases over rounds, consistent with prior work (Mordo et al., 2025a). In addition, increasing the number of RA agents generally leads to higher minimum inter-document similarity, contrasting with the single RA agent scenario (Figure 8a), where the RA agent modifies its documents more extensively than NA agents. This suggests that alternative modification strategies emerge when multiple RA agents compete.

Figure 10a shows the similarity between consecutive documents of the RA agents across settings. In all settings, similarity increases over rounds, consistent with prior work on ranking competitions between LLMs (Mordo et al., 2025a). No clear differences are observed between settings. A possible explanation is that the presence of multiple RA agents stabilizes document modification strategies across settings. Figure 10b presents the ranking scores across rounds, which exhibit a slight upward trend without statistically significant differences between settings.

Overall, these results indicate that introducing multiple RA agents influences document modification dynamics, increasing similarity between ranked documents while maintaining consistent ranking performance across rounds.

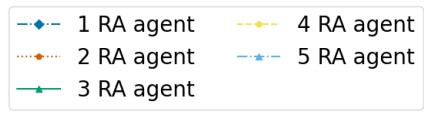

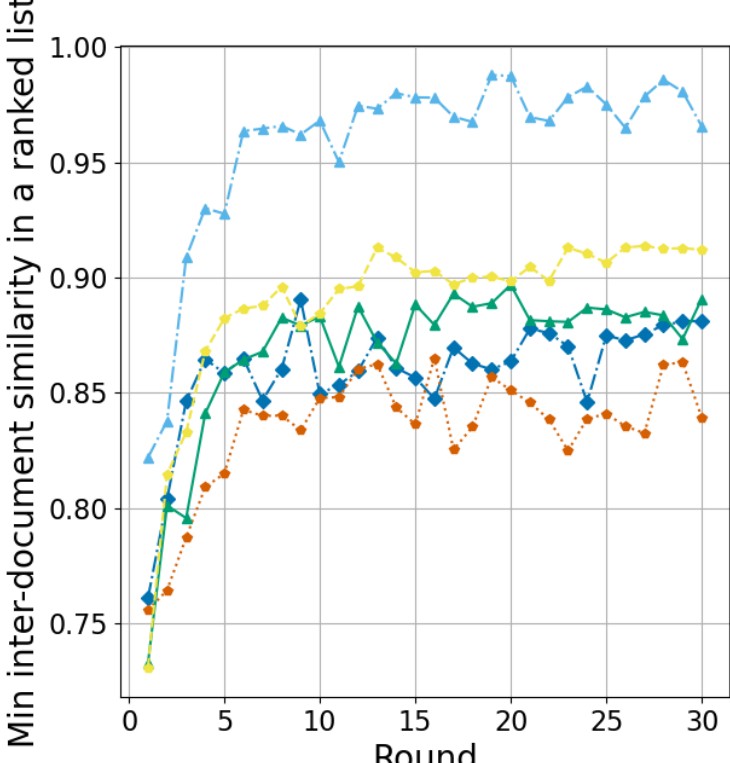

Figure 9: Comparison of the average minimum inter-document similarity in a ranked list (averaged across rounds) across settings with varying numbers of RA agents. In each setting, five agents compete: the number of RA agents ranges from one to five, and the remaining agents are NA agents. Each setting with $j$ RA agents is abbreviated as $j$ **RA agent**.

Table 10: Mean relevance judgment per configuration, player, and round. $r$ denotes statistical significance difference between rounds (01 vs.30) for the same agent and setting. $p$ denotes statistical significance difference between the NA and RA agents at the same round and setting. **Mean Rel.** is the mean relevance of the documents in the respective configuration. $\kappa$ is the inter-annotator agreement rates (free-marginal multi-rater Kappa) of the relevance judgment.

| Configuration | NA agent | | RA agent | | Mean Rel. / $\kappa$ |
|---|---|---|---|---|---|
| | Round 01 | Round 30 | Round 01 | Round 30 | |
| LSW+SG+He | 1.83 | $2.80^r$ | $2.73^p$ | 2.97 | 2.58 / 79% |
| LSW+DG+Ho | 1.83 | 2.30 | $2.73^p$ | 2.23 | 2.27 / 54% |
| LSW+DG+He | 1.83 | $2.77^r$ | $2.73^p$ | 2.80 | 2.53 / 76% |

## K  RELEVANCE JUDGMENTS

We annotated the datasets corresponding to competitions with Mistral agent: LSW+SG+He, LSW+DG+Ho, LSW+DG+He. Each document was judged for binary relevance to a query by three crowd workers (English speakers) on the Connect platform via CloudResearch (noa, 2024). We

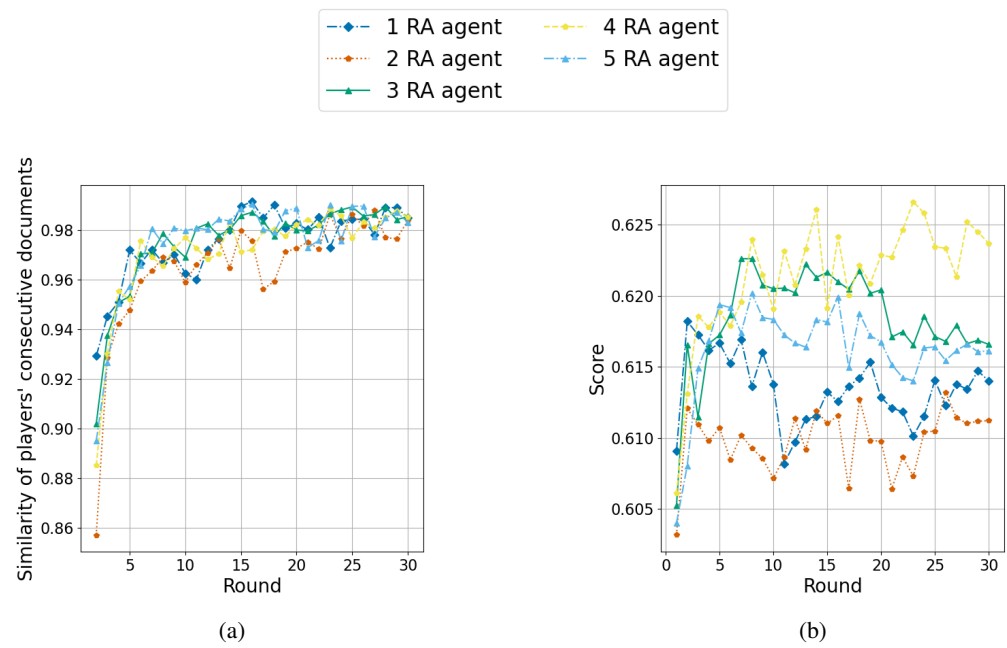

(a)                                                  (b)

Figure 10: Comparison of the RA agent and NA agents across settings with varying numbers of RA agents. In each setting, five agents compete: the number of RA agents ranges from one to five, and the remaining agents are NA agents. Each setting with $j$ RA agents is abbreviated as $j$ **RA agent**. We evaluate the following measures: (a) the average similarity (over queries) between consecutive rounds $(i, i + 1)$, and (b) the average ranking score over rounds (averaged over queries).

adopted the annotation guidelines from MS MARCO (Payal Bajaj et al., 2016; Craswell et al., 2025). The final relevance grade was defined as the number of annotators who marked it as relevant.

Due to budget limitations, we annotated only the RA agent and one (arbitrarily chosen) NA agent for rounds 1 and 30, enabling us to analyze the effect of the alignment process (i.e documents in round 1) and the competition dynamics (round 30). The inter-annotator agreement, measured with the free-marginal multi-rater Kappa statistic (Fleiss, 1971). The kappa agreement for the relevance judgments ranged between $54\%$–$79\%$.

We observe a clear distinction between the RA and the NA agents at the beginning of the competition. In round 1, the RA agent produces documents with higher average relevance ($2.73$ for RA agents in all three settings vs. $1.83$ for the NA agent), which we attribute to the alignment process during training that directly optimizes for ranker-preferred modifications. By round 30, however, this advantage diminishes, reflecting the herding effect whereby all agents progressively adapt toward the same high-relevance regions of the document space. When analyzing results per agent, we find that participation in the competition improves the relevance of the NA agent. For the RA agent a minor improvement was observed for the He settings.

## L   DECLARATION OF GENERATIVE AI USAGE IN THE WRITING PROCESS

We used an LLM (OpenAI's GPT-5) as a general-purpose writing assistant to improve the clarity and style of the paper. Its role was limited to language refinement and formatting support; all research ideas, methods, experiments, and analyses were carried out by the authors.

