# OpenReview forum: "RLRF: Competitive Search Agent Design via Reinforcement Learning from Ranker Feedback"
_ICLR.cc/2026/Conference — Submitted to ICLR 2026_

### Official Review · Reviewer_MhmN · 2025-10-27

**Soundness:** 3
**Presentation:** 3
**Contribution:** 2
**Rating:** 4
**Confidence:** 4

**Summary:**

This paper proposes a framework, RLRF, for training LLM agents to engage in competitive search—a task where agents strategically edit documents to achieve higher search rankings. The method involves simulating a multi-agent competitive environment to generate a preference dataset, which is then used to fine-tune an agent using DPO. The experiments demonstrate that this dynamically trained agent (RA) outperforms prompt-based baselines and shows an ability to generalize to unseen rankers. The authors also report that their agent, despite being trained solely on a ranking objective, exhibits better content faithfulness than the baselines.

**Strengths:**

1.	Clear Problem Formulation and Comprehensive Empirical Validation: The paper addresses a timely and interesting problem by formalizing strategic content generation as a learnable, multi-agent task. The experimental setup is comprehensive, providing a solid benchmark for future work in this area.

2.	Identifies Interesting Emergent Phenomena: The study successfully identifies two non-obvious and valuable phenomena: (1) the agent's emergent tendency to preserve content faithfulness despite a purely competitive objective, and (2) the asymmetric nature of knowledge transfer between different ranking functions. These findings could inspire follow-up research.

**Weaknesses:**

1.	Insufficient Conceptual and Methodological Novelty: The primary concern lie in the paper's lack of significant novelty. The core idea—applying a reinforcement learning-style algorithm to a game-theoretic problem—is a well-established paradigm. The methodology is a straightforward combination of existing techniques: using multi-agent simulation (akin to self-play) to generate data, followed by a standard preference alignment algorithm (DPO). While this "recipe" is effective, it does not introduce a novel algorithmic component, or a fundamentally new way of thinking about the problem. The central hypothesis that training on dynamic, competitive data (DG) is superior to training on static, isolated data (SG) is also obvious and confirms an existing intuition rather than revealing a new insight.

2.	Superficial Analysis of Key Findings: While the paper reports interesting emergent behaviors (faithfulness and asymmetric transfer), it fails to provide a deep investigation into mechanism behind these phenomena. For instance, explaining the emergent faithfulness could lead to profound insights about the implicit biases of modern neural rankers or the nature of strategic alignment. Without this deeper analysis, the findings remain intriguing observations. I recommend the authors could explore more on this aspect.

3.	Unaddressed Sim-to-Real Gap: The entire evaluation is conducted in a highly idealized simulation that abstracts away the most difficult aspects of the real-world problem: non-stationary and adversarial rankers, black-box algorithms, and sparse, delayed, and noisy feedback. The paper's claims of robustness are based on transferring between fixed, known-type rankers, which does not adequately address the challenges of a constantly evolving, real-world search ecosystem. A more critical discussion of these limitations is needed to properly contextualize the work's practical applicability.

**Questions:**

See Weaknesses

---

> ### Author Response · Authors · 2025-11-16
>
> W1: addressed in the comment "The novelty of the paper"
>
> W2: we thank the reviewer for this thoughtful comment. We agree that the emergent behaviors - particularly faithfulness and asymmetric transfer - deserve deeper investigation. We have conducted preliminary analyses aimed at explaining these mechanisms. For example, we observed that model size correlates positively with faithfulness (see Appendix F); recall the instruction to maintain faithfulness appears explicitly in the prompt to the agent. Understanding why larger models adhere more strongly to this instruction, and why the RL-aligned agent is slightly more faithful than the non-aligned agent, remains an open research question. We believe this touches on fundamental issues regarding implicit biases of modern LLM-based rankers and the nature of strategic alignment.
> Additionally, as noted in the Conclusion, we did not optimize the RL agent for faithfulness; the increased faithfulness emerged as a by-product rather than an explicit objective. Investigating the inherent trade-off between ranking promotion and faithfulness - and exploring optimization criteria that explicitly reward faithfulness - is an important direction for future work.
> Regarding asymmetric transfer, we elaborate in Comment 6VnK:Q1 on our effort for understanding this phenomenon in more depth. We view both behaviors as promising avenues for further study and expect that deeper analysis could yield meaningful theoretical insights.
>
> W3: addressed in the comment "Limitations of real world deployment"

---

### Official Review · Reviewer_AurJ · 2025-10-31

**Soundness:** 2
**Presentation:** 2
**Contribution:** 2
**Rating:** 4
**Confidence:** 2

**Summary:**

The paper studies the problem of editing documents via LLMs to sequentially improve their ranking in a search engine given a query, a setting defined as "competitive search". They propose to use the Ranking Feedback (RF) to finetune agents for the task, based on past rankings. Concretely, at each round, the highest and lowest performing document edits are stored and used to finetune the LLMs via the DPO algorithm. Experiments are performed in multi-agent simulations (where multiple agents compete for the highest rank) and show that agents aligned with RF outperform other baseline prompt-based approaches.

**Strengths:**

The paper formalizes a novel problem in information retrieval, and shows Reinforcement Learning from Ranking Feedback (RLRF) to be an effective algorithm to maximize ranking, with a clear hedge over simpler prompting baselines.

The experiments are sound and the RLRF agents seem able to generalize also to unseen ranking functions, making it more realistic for real-world scenarios where the ranking functions are unknown.

**Weaknesses:**

The following are the main weaknesses of the paper:

- The paper lacks novelty in terms of methods: the used approach is quite standard in the LLM literature: collect documents preference pairs --in this case using the ranking function-- and finetune the model via the DPO algorithm.

- The aligned agents are evaluated after DPO finetuning was completed. I think it would be interesting to see the evolution of their performance as a function of the funetuning steps. In particular, how much data (i.e. ranked document pairs) are required for them to outperform the other baselines?

Unfortunately, I cannot judge the relevance of the problem and whether it is of interest to the ICLR community.

**Questions:**

See weaknesses.

---

> ### Author Response · Authors · 2025-11-16
>
> W1: addressed in the comment: "The novelty of the paper"
>
> W2: we thank the reviewer for this important point. In practical deployments, there is indeed a real tradeoff between the amount of ranked-pair data used for DPO finetuning and the resulting performance of the aligned agents. We fully agree that analyzing performance as a function of finetuning steps - and identifying how much data is required for RA agents to surpass the baselines - is both important and informative. We are currently conducting this analysis, and due to resource constraints, it is still in progress. We hope to report the corresponding results in the coming days.
>
> W3: addressed in the comment: "Relevance to the ICLR community"

---

> > ### Author Response · Authors · 2025-11-24
> >
> > W2: we provided additional analysis of the performance as a function of the number of queries used for training in the comment "additional results" above.

---

### Official Review · Reviewer_iycE · 2025-11-01

**Soundness:** 3
**Presentation:** 3
**Contribution:** 3
**Rating:** 6
**Confidence:** 3

**Summary:**

This paper introduces Reinforcement Learning from Ranker Feedback (RLRF), a novel framework for training large language models (LLMs) as strategic agents in competitive search environments. The agents optimize document content to improve rankings against competitors by learning from preference datasets derived from ranking competitions. The authors propose two data generation methods: Static Generation (SG) (independent document modifications) and Dynamic Generation (DG) (simulated multi-agent competitions). Experiments using the LEMSS simulator show that RL-aligned agents (RA agents) consistently outperform non-aligned prompt-based agents (NA agents) across diverse ranking functions (e.g., E5, Contriever, BM25). RA agents also generalize to unseen rankers and adapt to strategic opponents. Key contributions include formalizing competitive search as a learning problem, demonstrating RLRF’s effectiveness, and highlighting transfer learning capabilities.

**Strengths:**

- Originality: Novel formulation of competitive search as an RL problem, with DG simulating strategic opponent adaptation.
- Quality: Rigorous experiments across 4 LLMs, 4 rankers, and homogeneous/heterogeneous settings. RA agents (e.g., Mistral+DG+LSW) achieve up to 75% win-rate (Ho) and 60% (He).
- Clarity: Clear problem definition (Section 3) and accessible methodology (e.g., DPO for alignment).
- Significance: Demonstrates transfer learning to unseen rankers (Table 2), with asymmetric generalization (e.g., Contriever-trained agents generalize better than E5-trained).

**Weaknesses:**

- Synthetic Limitations: Experiments rely on simulated competitions (LEMSS) and MS MARCO data. Real-world deployment (e.g., dynamic user queries) is unexplored.
- Scalability: Training RA agents requires multi-round simulations (450 games × 30 rounds), but computational costs are not quantified.
- Strategy Stability: Convergence analysis (Appendix G) shows agents gravitate toward similar documents over time, but robustness to adversarial opponents (e.g., non-LLM agents) is untested.

**Questions:**

See weakness.

---

> ### Author Response · Authors · 2025-11-16
>
> W1: addressed in the comment above: "Limitations of real world deployment"
>
> W2: We thank the reviewer for this important comment and will include a concise discussion in the revised paper. First, we emphasize that training RA agents requires simulating a competition consisting of at least k + 1 rounds, where k denotes the history depth provided to the agent as context input during the competition. Consequently, for the LSW and PAW prompts, three and four rounds are required, respectively. To achieve this, one can employ a single LLM sharded across two A100 40GB GPUs with a batch size of 16, resulting in 85 and 113 inferences per competitor, respectively; the number of inferences are the result of: dataset_size * history_depth / batch_size.
>
> W3: We appreciate this insight. Indeed, in strategic settings the behavior of agents is inherently dependent on their opponents, and the convergence patterns we observe reflect the well-known game-theoretic phenomenon wherein mimicking the current winner becomes an equilibrium strategy. While Appendix G demonstrates that our RA agents tend to converge toward similar documents over time, we agree that evaluating their robustness against adversarial or non-LLM opponents is an important open question. Understanding how RA agents perform when facing strategically designed adversaries is a valuable direction for future research, and we plan to explore this in subsequent work.

---

### Official Review · Reviewer_6VnK · 2025-11-03

**Soundness:** 3
**Presentation:** 3
**Contribution:** 2
**Rating:** 4
**Confidence:** 2

**Summary:**

This paper addresses the problem of "competitive search," a dynamic setting where document publishers strategically modify their content to improve their ranking on search engines. Recognizing that publishers increasingly leverage Large Language Models (LLMs) for this task, the authors introduce "Reinforcement Learning from Ranker Feedback" (RLRF), a novel framework for training these LLM-based publisher agents. The methodology involves training an LLM—termed an "RL-aligned agent" or "RA agent" —using preference datasets. These datasets are synthetically generated from simulated ranking competitions, thereby encoding signals from the ranker's outputs (i.e., the rankings). The agent alignment is performed at training time, notably using Direct Preference Optimization (DPO) , which allows the finalized RA agent to operate at test time via simple prompting without further optimization.

**Strengths:**

The primary novelty is the RLRF framework itself—the conceptual reframing of alignment-RL for a competitive, game-theoretic task. The agent trained on SG learns to hit a static target. The agent trained on DG learns to win a dynamic game. For any competitive, multi-agent domain, training agents on static preference data is fundamentally insufficient. One must incorporate multi-agent simulation (like DG) into the training loop to learn the "meta-game."

**Weaknesses:**

1. The paper dismisses Proximal Policy Optimization (PPO) as "less stable"  and adopts DPO. While DPO is a strong and modern choice, this justification is brief. A more detailed explanation or a preliminary experiment comparing them would have strengthened this methodological choice.
2. The paper's claim from Figure 2 (that RLRF improves faithfulness) is true only when controlling for model size. The paper fails to address the critical question: "How does an 8B-RA agent compare to a 70B-NA agent?" This weakness badly muddies the conclusions about faithfulness. It is plausible that the primary benefit of RLRF is rank promotion, and faithfulness is almost entirely a function of model scale.

**Questions:**

1. Could you please elaborate on your hypothesis for the "asymmetric" transfer learning observed in Table 2? Why does Contriever appear to be a "better teacher" than E5-unsupervised? Does this imply that the choice of generation ranker is a critical, first-class component of the RLRF framework that requires its own line of research?
2. could you provide a brief sensitivity analysis for a key parameter, such as the DPO beta? This would help confirm the "robustness" interpretation and allay fears that the results are specific to the chosen default.

---

> ### Author Response · Authors · 2025-11-16
>
> W1: We appreciate the reviewer’s comment. Our choice of DPO was motivated by its greater stability and lower optimization complexity in preference-based alignment settings. Actor-critic methods such as PPO require simultaneous learning of a value function and a policy, which introduces known sources of instability, and PPO further requires a separate reward model, rollout generation, and KL tuning. In contrast, DPO implicitly optimizes the same constrained RL objective using a supervised-style loss, avoiding critic learning entirely and resulting in a method that is markedly easier to tune and computationally lighter - an important consideration in our multi-agent simulation setup.
> We also note that our goal in this work is to provide a proof of concept demonstrating that RL-style preference optimization can improve content generation in ranking competitions, rather than to identify the globally optimal RL algorithm. Given our limited computational resources, we prioritized a stable and lightweight method, and we acknowledge that in some settings carefully tuned PPO can outperform DPO. We will clarify these motivations and trade-offs in the revised paper.
> In the revised paper, we will expand Section 3 with a more detailed and well-supported discussion of the trade-offs between PPO and DPO - including stability considerations, optimization complexity, and resource.
> refs:
> arxiv.org/abs/2305.18290
> arxiv.org/pdf/2004.14288
> arxiv.org/pdf/1506.02438
>
> W2: We thank the reviewer for the comment. Our experiments indeed focused on lightweight models, where resource constraints allowed systematic study across multiple rankers and learning settings. Within this scope, we consistently observed that RLRF-trained (RA) agents are more faithful than non-aligned (NA) agents, even though RLRF is not directly optimized for faithfulness - an interesting emergent result. We agree that a full comparison between lightweight and large models would provide additional insight. In the paper, we included results for larger NA models as inspiration, showing that faithfulness tends to increase with model scale. We hypothesize that larger RA agents would exhibit the same trend, as larger LLMs typically follow the instructional prompts given to the LLM to a larger extent [In the prompt to the LLMs we ask all the agents to “...Ensure the edited document is very similar to the candidate document…”]. However, due to computational limits, we could not currently validate this experimentally. In the revised version of the paper, we will:
> 1.	Clarify that the faithfulness analysis is scoped to lightweight models.
> 2.	Emphasize that RA > NA in faithfulness holds consistently within that regime.
> 3.	Add a short discussion on the hypothesized scaling behavior of RA agents and a note about computational limitations.
>
> Q1:We thank the reviewer for this insightful question. As our results indicate, using Contriever as the training ranker leads to better performance when E5-unsupervised serves as the competition ranker, compared to the reverse configuration - suggesting an asymmetric transfer between training and evaluation rankers. To explore this phenomenon, we conducted a comprehensive analysis: (1) measuring the similarity between the ranked document lists induced by the training ranker and those produced by the evaluation rankers; (2) comparing differences in document-query similarity scores across rankers; and (3) applying these analyses both to the generated documents during training and to the selected / rejected document pairs used by the DPO alignment process. Despite these efforts, we did not identify a definitive mechanism explaining the phenomenon. We believe this asymmetry highlights an underexplored aspect of generation - ranker interactions, and we recommend initiating a dedicated line of research to better understand and characterize it.
> As proposed by the reviewer, we will discuss in the revised paper on the importance of ranker selection for training as a crucial element in the RLRF framework.
>
> Q2: We thank the reviewer for the helpful suggestion. Our experiments intentionally focused on robustness at inference time - varying LLM parameters such as temperature and environmental factors such as the number of opponents and the ranker type - rather than on hyperparameter tuning during training. We agree that a sensitivity analysis of the DPO β parameter is valuable, and we are currently running additional experiments to assess its effect. We note that our primary goal was to provide a proof of concept showing that RL-style preference optimization can improve agent performance in ranking competitions; even without tuning β or other training hyperparameters, our RA agents already outperform strong baselines. As discussed in Appendix C.2, we expect that effective hyperparameter tuning, including β, would further improve performance. We will incorporate a brief β-sensitivity analysis, subject to resource limits, in the revised paper.

---

> ### Author Response · Authors · 2025-11-24
>
> Q2: We provided a sensitivity analysis of DPO β in the comment "additional results" above.

---

### Author Response · Authors · 2025-11-16
**Disclaimer**

We thank all the reviewers for their insightful and constructive feedback. In this rebuttal, we address all points raised. Due to limited computational resources, several additional analyses are still running, and we expect to report the results and integrate them into the revised paper by the end of the discussion period. Specifically, we are currently conducting:
(a) a sample-complexity analysis examining performance as a function of the number of fine-tuning steps (as raised in AurJ:W2), and
(b) a sensitivity analysis of the RA agents with respect to different DPO β values (as suggested in 6VnK:Q2).

---

### Author Response · Authors · 2025-11-16
**Limitations of real world deployment**

While we agree that there is a non-trivial gap between real-world scenarios and our experimental environment, which simulates competitive search using MS MACRO data, we would like to point out that similar simulation-based approaches have been shown to be useful in various contexts, both within and beyond the scope of information retrieval (IR).

In the context of search and recommender systems, several recent works have utilized similar simulation-based approaches to imitate real-world system behavior. For instance, Ye et al. (2025) developed an LLM-driven simulation of the recommender ecosystem to improve the evaluation of long-term creator dynamics; Nachimovsky et al. (2025) adopted a similar approach to study the effect of heterogeneity of AI agents in different roles (publisher agent, query formulator, and ranker) on various aspects of the search ecosystem. Beyond the IR context (but still within the scope of ML), recent work by Shapira et al. (2024, 2025) utilized simulation-based approaches to generate synthetic data that mimics human behavior in sequential decision-making scenarios, and used it to predict human choice behavior in persuasion games.

Relying on this extensive line of research that demonstrates the relevance of simulations (and particularly, LLM-based simulations) to real-world scenarios, we view our work as a significant step towards developing effective mechanisms for competitive search agent design, which could potentially be integrated and tested in non-simulated environments, as part of a future research effort.
In addition, exploring settings where user queries evolve over time is an important direction for future work. Conceptually, this follows the natural progression in the IR community - early work typically focused on single-query scenarios and only later moved to multi-query and dynamic-query settings. We believe a similar stepwise development is appropriate here: first establish and validate RLRF in the controlled, single-query / simulated-competition regime (this paper), and then extend it to increasingly realistic and temporally evolving query distributions in subsequent work. This staged approach both clarifies causal mechanisms and ensures the community can build reliable methods incrementally.

References:

Nachimovsky, H., Tennenholtz, M., & Kurland, O. (2025). A Multi-Agent Perspective on Modern Information Retrieval. arXiv preprint arXiv:2502.14796.

Shapira, E., Madmon, O., Reichart, R., & Tennenholtz, M. (2024). Can llms replace economic choice prediction labs? the case of language-based persuasion games. arXiv preprint arXiv:2401.17435.

Shapira, E., Madmon, O., Apel, R., Tennenholtz, M., & Reichart, R. (2025). Human choice prediction in language-based persuasion games: Simulation-based off-policy evaluation. TACL 2025.

Ye, X., Xu, C., Sun, Z., Xu, J., Wang, G., Dong, Z., & Wen, J. R. LLM-empowered creator simulation for long-term evaluation of recommender systems under information asymmetry. SIGIR 2025.

---

### Author Response · Authors · 2025-11-16
**The novelty of the paper**

Our work is the first, to the best of our knowledge, to align (via RL) content generation (and more generally document creation/modification) with ranking preferences induced by an undisclosed ranker while accounting for content faithfulness: a fundamental challenge in competitive search from an agent perspective. While we acknowledge that our work does not introduce algorithmic novelty, we emphasize its significant conceptual and applicative novelty across several dimensions:
(1) Competitive search as a learning problem.
Prior work on competitive search primarily analyzes the ranker’s perspective: how the ranking mechanism influences publishers' behavior. In contrast, we study the agent’s perspective: how LLM-based publishers can learn to generate documents under competition. To our knowledge, this is the first formulation of competitive search as a learning task, enabling systematic evaluation of RL-based alignment techniques for strategic content generation, going beyond prompt engineering as in Bardas et al. (2025) [full citation in the paper]. This perspective connects competitive search to broader ML settings involving dynamic, incentive-driven evaluation.
(2) Search competition ranking outcomes as alignment feedback.
We introduce the use of an unknown ranking algorithm’s output as feedback for alignment. While self-play and RL-based fine-tuning are well-known, existing “AI feedback” setups (e.g., RLAIF) typically use AI proxies to replace human preferences. Our goal is different: to align LLMs with an unknown ranker, through rankings that depend jointly on the input query and on the model’s own competing, self-generated documents, while the downstream competition remains unobserved. This constitutes alignment to an objective defined by competitive interactions, a setting not explored in prior alignment studies.
(3) Incentive-aware synthetic data generation.
 We decompose two core components of the task: algorithmic alignment to the ranker’s preference and strategic adaptation to the competitive landscape. We operationalize this distinction via two data generation schemes, Static Generation (SG) and Dynamic Generation (DG), showing that DG, which simulates real competition, leads to significantly higher performance gains. This finding formalizes the importance of strategic synthetic data for learning in competitive environments, extending similar insights recently observed in human-choice prediction tasks (e.g., Shapira et al., 2024 [full citation in the paper]).
(4) In this sense, our methodology can be viewed as synthetic data generation for settings in which the agent is not only uncertain about the game itself (e.g., the ranker), but also about its opponents, the mapping from strategies (corpora) to utilities, and even the structure of the strategy space - here instantiated as free-text inputs with an unknown embedding geometry. This makes the problem substantially more challenging than standard stylized game-theoretic environments, where self-play is typically applied. Moreover, whereas self-play is generally used to iteratively improve a policy within a fixed game, our approach focuses on simulating families of synthetic games in order to produce data that trains a model to perform robustly in new, unseen strategic environments.
Lastly, while these contributions stand on their own, they also complement ongoing algorithmic work in RL-based alignment. By highlighting the unique challenges of alignment in competitive search - such as inter-agent dependencies and non-stationary feedback - they open the door to developing new alignment methods that explicitly account for strategic and dynamic environments.

---

### Author Response · Authors · 2025-11-16
**Relevance to the ICLR community**

We would like to emphasize the strong relevance of our work to the ICLR community. Technically, our study builds directly on core ML methodologies - LLM fine-tuning, RL-based alignment, preference optimization, and self-play - and adapts them to a new and practically important feedback regime. The proposed RLRF framework extends the family of RLHF/RLAIF approaches to settings where the feedback signal is an algorithmic ranking rather than human annotations. This introduces unique challenges, particularly the need for opponent-strategy adaptation: because agents condition on competition history and previous ranked lists during training, they must learn behaviors that remain effective under different strategic responses and player interactions. In this sense, our setting provides a natural testbed for studying alignment and optimization methods in multi-agent competitive environments.
From an application standpoint, our setting of competitive search lies at the intersection of information retrieval, recommender systems, and algorithmic game theory, which are long-standing and central research areas within ML, as evidenced by the high number of papers in those fields consistently being published at top-tier AI/ML conferences like ICLR/NeurIPS/AAAI (e.g., Ben-Porat et al. 2018, 2019, Yao et al. 2023, 2024, Nachimovsky et al. 2025, Madmon et al. 2025, and more). The ability to train LLM-based agents that adapt strategically to ranking algorithms has direct implications for applied ML domains such as content optimization and multi-agent learning. We thus view our contribution as both methodologically grounded and directly relevant to active areas of ML research and practice.

References:

Ben-Porat, Omer, and Moshe Tennenholtz. "A game-theoretic approach to recommendation systems with strategic content providers." Advances in Neural Information Processing Systems 31 (2018).

Ben-Porat, Omer, Itay Rosenberg, and Moshe Tennenholtz. "Convergence of learning dynamics in information retrieval games." Proceedings of the AAAI Conference on Artificial Intelligence. Vol. 33. No. 01. 2019.

Omer Madmon, Idan Pipano, Itamar Reinman, and Moshe Tennenholtz. On the convergence of no-regret dynamics in information retrieval games with proportional ranking functions. In The Thirteenth International Conference on Learning Representations, 2025.

Haya Nachimovsky and Moshe Tennenholtz. On the power of strategic corpus enrichment in content creation games. In Proceedings of the AAAI Conference on Artificial Intelligence, volume 39, pages 14019–14026, 2025.

Yao, F., Li, C., Nekipelov, D., Wang, H., & Xu, H. (2023). How Bad is Top-$ K $ Recommendation under Competing Content Creators?. In International Conference on Machine Learning (pp. 39674-39701). PMLR.

Yao, F., Liao, Y., Liu, J., Nie, S., Wang, Q., Xu, H., & Wang, H. (2024). Unveiling user satisfaction and creator productivity trade-offs in recommendation platforms. Advances in Neural Information Processing Systems, 37, 86958-86984.

---

### Author Response · Authors · 2025-11-24
**Additional results**

We are pleased to report that the additional analyses requested by the reviewers have now been completed:
(a) Sample-complexity analysis: We trained Mistral DG (Dynamic Generation) agents using 50 to 450 queries in steps of 50. Every agent was evaluated in a homogenous setup (i.e., four competitors, all NA Mistral agents). The RA agent’s win rate ranges from 0.51 to 0.82, increasing steadily with the number of fine-tuning samples and showing a tendency to converge as the sample size grows. This demonstrates that RA agents begin to outperform baselines NA agents with a moderate number of training pairs.

(b) DPO β sensitivity analysis: In this experiment, we used the Mistral agent trained with dynamic generation (DG) datasets, evaluated in a homogenous evaluation setting (all opponents are NA Mistral). The RA agent’s win rate ranges from 0.50 to 0.75 and is statistically significantly higher than that of the NA agent across all tested β values. These results indicate that RA performance exhibits low sensitivity to variations in β, confirming that our findings do not rely on the default choice of β.

Overall, these analyses strengthen our conclusions, showing that RA agents achieve strong performance with a moderate number of training pairs and maintain robust advantages over NA agents across reasonable choices of β. We will include these results in the revised version of the paper.

---

### Author Response · Authors · 2025-12-03
**Summary of the rebuttal**

Dear AC,

Here is a concise summary of what was addressed in the rebuttal:

Faithfulness analysis: Clarified model-size effects, emphasized findings within lightweight models, and explained why RA agents exhibit slightly higher faithfulness.

DPO vs. PPO justification: Added a detailed, well-supported discussion of why DPO was chosen (stability, simplicity, resource constraints, and proof-of-concept nature).

DPO β sensitivity: Ran additional experiments showing RA performance is not sensitive across β values.

Sample-complexity analysis: Added experiments showing RA agents' performance improves with more fine-tuning samples and achieves meaningful gains with modest data.

Asymmetric transfer: Described the Conducted analyses (rank-list similarity, scoring differences, DPO pair behavior) and reported that no definitive mechanism was found; identified this as a direction for new research.

Scalability / multi-round training cost: Clarified computational requirements for training RA agents with multi-round prompts.

Strategy stability / adversarial opponents: Discussed equilibrium considerations and highlighted adversarial-opponent robustness as future work.

Novelty clarification: Strengthened the positioning of RLRF as extending RLHF/RLAIF to ranking-feedback environments with opponent-dependent, competition-history-aware learning. We added as well a discussion about the relevance of the paper to the ICLR community.

Additional minor clarifications: Addressed multiple reviewer-specific technical questions throughout the rebuttal.

Contextualizing our simulated setting within a well-established line of IR/ML simulation research, clarifying its role as a foundational step toward real-world deployment, and outlining a clear future path toward more realistic, evolving-query environments.

---

### Meta-Review · Area_Chair_4Sm1 · 2026-01-08

**Summary:**

The paper proposes a framework that trains LLMs to optimize competitive content using preference data derived from ranking competitions without human-authored data.


+ novelty concerns: the method is a standard recipe (simulation + preference pairs + DPO) with limited algorithmic novelty.

+ choice between DPO and PPO needs to be justified

+ Results might depend on a specific β choice.

+ Unclear how much ranked data is needed for RA agents to outperform baselines.

+ Faithfulness improvements may be confounded by model size; unclear whether RLRF truly helps beyond scale effects

+ unclear why does training on some rankers (e.g., Contriever) generalize better than others?

+ robustness to adversarial or non-LLM opponents is unclear.

**Reviewer Concerns:**

Outstanding concerns:

+ Mechanisms behind emergent faithfulness are unclear; only preliminary analysis and hypotheses provided.

+ Asymmetric transfer between rankers was investigated empirically but unclear conclusion

+ Robustness to adversarial or non-LLM opponents remains unclear

+ Gap with realistic scenarios like dynamic queries, evolving/black-box rankers

**Reviewer Scores:**

The reviewers are mostly negative, and I doubt the discussion would have significantly changed the situation.

---

### Decision · Program_Chairs · 2026-01-26

Reject